# Noise-Aware Differentially Private Regression via Meta-Learning

**Ossi Räisä**[*]
University of Helsinki
ossi.raisa@helsinki.fi

**Stratis Markou**[*]
University of Cambridge
em626@cam.ac.uk

**Matthew Ashman**
University of Cambridge
mca39@cam.ac.uk

**Wessel P. Bruinsma**
Microsoft Research AI for Science
wessel.p.bruinsma@gmail.com

**Marlon Tobaben**
University of Helsinki
marlon.tobaben@helsinki.fi

**Antti Honkela**
University of Helsinki
antti.honkela@helsinki.fi

**Richard E. Turner**
University of Cambridge
ret26@cam.ac.uk

## Abstract

Many high-stakes applications require machine learning models that protect user privacy and provide well-calibrated, accurate predictions. While Differential Privacy (DP) is the gold standard for protecting user privacy, standard DP mechanisms typically significantly impair performance. One approach to mitigating this issue is pre-training models on simulated data before DP learning on the private data. In this work we go a step further, using simulated data to train a meta-learning model that combines the Convolutional Conditional Neural Process (ConvCNP) with an improved functional DP mechanism of Hall et al. [2013] yielding the DPConvCNP. DPConvCNP learns from simulated data how to map private data to a DP predictive model in one forward pass, and then provides accurate, well-calibrated predictions. We compare DPConvCNP with a DP Gaussian Process (GP) baseline with carefully tuned hyperparameters. The DPConvCNP outperforms the GP baseline, especially on non-Gaussian data, yet is much faster at test time and requires less tuning.

## 1 Introduction

Deep learning has achieved tremendous success across a range of domains, especially in settings where large datasets are publicly available. However, in many impactful applications such as healthcare, the data may contain sensitive information about users, whose privacy we want to protect. Differential Privacy [DP; Dwork et al., 2006] is the gold standard framework for protecting user privacy, as it provides strong guarantees on the privacy loss incurred on users participating in a dataset. However, enforcing DP often significantly impairs performance. A recently proposed method to mitigate this issue is to pre-train a model on non-private data, e.g. from a simulator [Tang et al., 2023], and then fine-tune it under DP on real private data [Yu et al., 2021, Li et al., 2022, De et al., 2022].

We go a step further and train a *meta-learning* model with a DP mechanism inside it (Figure 1). While supervised learning is about learning a mapping from inputs to outputs using a learning algorithm, in meta-learning we learn a learning algorithm directly from the data, by *meta-training*, enabling generalisation to new datasets during *meta-testing*. Our model is meta-trained on simulated datasets,

---

[*]Equal contribution.

38th Conference on Neural Information Processing Systems (NeurIPS 2024).

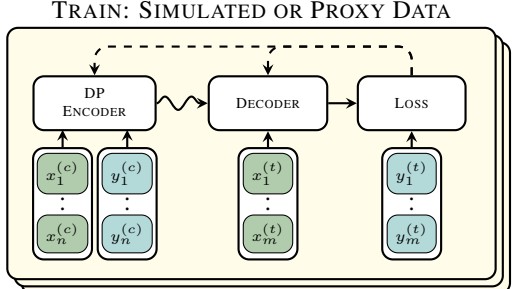 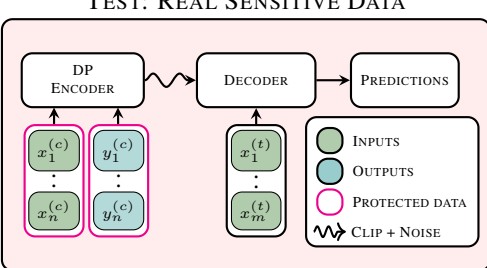

Figure 1: Meta-training (left) and meta-testing (right) using our method. We train a model on multiple tasks with non-private (simulated or proxy) data to predict on target $(t)$ points using the context $(c)$ points. Crucially, by including a DP mechanism, which clips and adds noise to the data *during training*, the parameter updates (dashed arrow) teach the model to make well-calibrated and accurate predictions in the presence of DP noise. At test time, we deploy the model on real data using the same mechanism, which protects the context set with DP guarantees.

each of which is split into a *context* and *target* set, learning to make predictions at the target inputs given the context set. At meta-test-time, the model takes a context set of real data, which is protected by the DP mechanism, and produces noise-aware predictions and accurate uncertainty estimates.

**Neural Processes.** Our method is based on neural processes [NPs; Garnelo et al., 2018a], a model which leverages the flexibility of neural networks to produce well-calibrated predictions in the meta-learning setting. The parameters of the NP are meta-trained to generalise to unseen datasets, while adapting to new contexts much faster than gradient-based fine-tuning alternatives [Finn et al., 2017].

**Convolutional NPs.** We focus on convolutional conditional NPs [ConvCNPs; Gordon et al., 2020], a type of NP that has remarkably strong performance in challenging regression problems. That is because the ConvCNP is translation equivariant [TE; Cohen and Welling, 2016], so its outputs change predictably whenever the input data are translated. This is an extremely useful inductive bias when modelling, for example, stationary data. The ConvCNP architecture also makes it natural to embed an especially effective DP mechanism inside it using the *functional mechanism* [Hall et al., 2013] to protect the privacy of the context set (Figure 1). We call the resulting model the DPConvCNP.

**Training with a DP mechanism.** A crucial aspect of our approach is training the DPConvCNP on non-sensitive data *with the DP mechanism in the training loop*. The mechanism involves clipping and adding noise, so applying it only during testing would create a mismatch between training and testing. Training with the mechanism eliminates this mismatch, ensuring calibrated predictions (Figure 2).

**Overview of contributions.** In summary, our main contributions in this work are as follows.

1. We introduce the DPConvCNP, a meta-learning model which extends the ConvCNP using the functional DP mechanism [Hall et al., 2013]. The model is meta-trained with the mechanism in place, learning to make calibrated predictions from the context data under DP.

2. We improve upon the functional mechanism of Hall et al. [2013] by leveraging Gaussian DP theory [Dong et al., 2022], showing that context set privacy can be protected with smaller amounts of noise (at least 25% lower standard deviation in the settings considered in Figure 4). We incorporate these improvements into DPConvCNP, but note that they are also of interest in any use case of the functional mechanism.

3. We conduct a study on synthetic and sim-to-real tasks. Remarkably, even with relatively few context points (a few hundreds) and modest privacy budgets, the predictions of the DPConvCNP are surprisingly close to those of the non-DP optimal Bayes predictor. Further, we find that a single DPConvCNP can be trained to generalise across generative processes with different statistics and privacy budgets. We also evaluate the DPConvCNP by training it on synthetic data, and testing it on a real dataset in the small data regime. In all cases, the DPConvCNP produces well calibrated predictions, and is competitive with a carefully tuned DP Gaussian process baseline.

## 2   Related Work

Training deep learning models on public proxy datasets and then fine-tuning with DP on private data is becoming increasingly common in computer vision and natural language processing applications

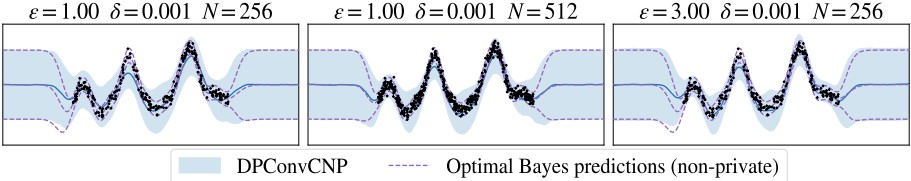

| $\varepsilon = 1.00$ $\delta = 0.001$ $N = 256$ | $\varepsilon = 1.00$ $\delta = 0.001$ $N = 512$ | $\varepsilon = 3.00$ $\delta = 0.001$ $N = 256$ |

DPConvCNP    - - - - Optimal Bayes predictions (non-private)

Figure 2: Training our proposed model with a DP mechanism inside it, enables the model to make accurate well-calibrated predictions, even for modest privacy budgets and dataset sizes. Here, the context data (black) are protected with different $(\epsilon, \delta)$ DP budgets as indicated. The model makes predictions (blue) that are remarkably close to the optimal (non-private) Bayes predictor.

[Yu et al., 2021, Li et al., 2022, De et al., 2022, Tobaben et al., 2023]. However, these approaches rely on the availability of very large non-sensitive datasets. Because these datasets would likely need to be scraped from the internet, it is unclear whether they are actually non-sensitive [Tramèr et al., 2024]. On the other hand, other approaches study meta-learning with DP during meta-training [Li et al., 2020, Zhou and Bassily, 2022], but do not enforce privacy guarantees at meta-test time.

Our approach fills a gap in the literature by enforcing privacy of the meta-test data with DP guarantees (see Figure 1), and using non-sensitive proxy data during meta-training. Unlike other approaches which rely on large fine tuning datasets, our method produces well-calibrated predictions even for relatively small datasets (a few hundred datapoints). In this respect, the work of Smith et al. [2018], who study Gaussian process (GP) regression under DP for the small data regime, is perhaps most similar to ours. However, Smith et al. [2018] enforce privacy constraints only with respect to the output variables and do not protect the input variables, whereas our approach protects both.

In terms of theory, there is fairly limited prior work on releasing functions with DP guarantees. Our method is based on the functional DP mechanism of Hall et al. [2013] which works by adding noise from a GP to a function to be released. This approach works especially well when the function lies in a reproducing kernel Hilbert space (RKHS), a property which we leverage in the DPConvCNP. We improve on the original functional mechanism by leveraging Gaussian DP theory of Dong et al. [2022]. In related work, Aldà and Rubinstein [2017] develop the Bernstein DP mechanism, which adds noise to the coefficients of the Bernstein polynomial of the released function, and Mirshani et al. [2019] generalise the functional mechanism beyond RKHSs. Jiang et al. [2023] derive Rényi differential privacy [RDP; Mironov, 2017] bounds for the mechanism of Hall et al. [2013].

## 3 Background

We start by laying the necessary background. In Section 3.1, we outline meta-learning and NPs, focusing on the ConvCNP. In Section 3.2 we introduce DP, and the functional mechanism of Hall et al. [2013]. We keep the discussion on DP lightweight, deferring technical details to Appendix A.

### 3.1 Meta-learning and Neural Processes

**Supervised learning.** Let $\mathcal{D}$ be the set of datasets consisting of $(x, y)$-pairs with $x \in \mathcal{X} \subset \mathbb{R}^d$ and $y \in \mathcal{Y} \subset \mathbb{R}$. The goal of supervised learning is to use a dataset $D \in \mathcal{D}$ to learn appropriate parameters $\theta$ for a conditional distribution $p(y|x, \theta)$, which maximise the predictive log-likelihood on unseen, randomly sampled test pairs $(x^*, y^*)$, i.e. $\mathcal{L}(\theta, (x^*, y^*)) = \log p(y^*|x^*, \theta)$. Let us denote the entire algorithm that performs learning, followed by prediction, by $\pi$, that is $\pi(x^*, D) = p(\cdot|x^*, \theta^*)$, where $\theta^* = \arg\max_\theta \mathcal{L}(r, D)$. Supervised learning is concerned with designing a hand-crafted $\pi$, e.g. picking an appropriate architecture and optimiser, which is trained on a single dataset $D$.

**Meta-learning.** Meta-learning can be regarded as supervised learning of the function $\pi$ itself. In this setting, $D$ is regarded as part of a single training example, which means that a meta-learning algorithm can handle different $D$ at test time. Concretely, in meta-learning, we have $\pi_{\theta,\phi}(x^*, D) = p(\cdot|x^*, \theta, r_\phi(D))$, where $r_\phi$ is now a function that produces task-specific parameters, adapted for $D$. The meta-training set now consists of a collection of datasets $(D_m)_{m=1}^M$, often referred to as *tasks*. Each task is partitioned into a context set $D^{(c)} = (\mathbf{x}^{(c)}, \mathbf{y}^{(c)})$ and a target set $D^{(t)} = (\mathbf{x}^{(t)}, \mathbf{y}^{(t)})$. We refer to $\mathbf{x}^{(c)}$ and $\mathbf{y}^{(c)}$ as the *context inputs and outputs* and to $\mathbf{x}^{(t)}$ and $\mathbf{y}^{(t)}$ as the *target inputs and outputs*. To meta-train a meta-learning model, we optimise its predictive log-likelihood, averaged over tasks, i.e. $\mathbb{E}_D[\mathcal{L}(\theta, \phi, D)] = \mathbb{E}_D[\log \pi_{\theta,\phi}(\mathbf{x}^{(t)}, D^{(c)})(\mathbf{y}^{(t)})]$. Meta-learning algorithms are broadly categorised into two groups, based on the choice of $r_\phi$ [Bronskill, 2020].

**Gradient based vs amortised meta-learning.** On one hand, gradient-based methods, such as MAML [Finn et al., 2017] and its variants (e.g. [Nichol et al., 2018]) rely on gradient-based fine-tuning at test time. Concretely, these let $r_\phi$ be a function that performs gradient-based optimisation. For such algorithms, we can enforce DP with respect to a meta-test time dataset by fine-tuning with a DP optimisation algorithm, such as DP-SGD [Abadi et al., 2016]. While generally effective, such approaches can require significant resources for fine-tuning at meta-test-time, as well as careful DP hyper-parameter tuning to work at all. On the other hand, there are amortised methods, such as neural processes [Garnelo et al., 2018a], prototypical networks [Snell et al., 2017], and matching networks [Vinyals et al., 2016], in which $r_\phi$ is a learnable function, such as a neural network. This approach has the advantage that it requires far less compute and memory at meta-test-time. In this work, we focus on neural processes (NPs), and show how $r_\phi$ can be augmented with a DP mechanism to make well calibrated predictions, while protecting the context data at meta test time.

**Neural Processes.** Neural processes (NPs) are a type of model which leverage the flexibility of neural networks to produce well calibrated predictions. A range of NP variants have been developed, including conditional NPs [CNPs; Garnelo et al., 2018a], latent-variable NPs [LNPs; Garnelo et al., 2018b], Gaussian NPs [GNPs; Markou et al., 2022], score-based NPs Dutordoir et al. [2023], and autoregressive NPs [Bruinsma et al., 2023]. In this work, we focus on CNPs because these are ideally suited for our purposes, but our framework can be extended to other variants. A CNP consists of an *encoder* $\texttt{enc}_\phi$, and a *decoder* $\texttt{dec}_\theta$. The encoder is a neural network which ingests a context set $D^{(c)} \in \mathcal{D}$ and outputs a representation $r$ in some representation space $\mathcal{R}$. Two concrete examples of such encoders are DeepSets [Zaheer et al., 2017] and SetConv layers [Gordon et al., 2020]. The decoder is another neural network, with parameters $\theta$, which takes the representation $r$ together with target inputs $\mathbf{x}^{(t)}$ and produces predictions for the corresponding $\mathbf{y}^{(t)}$. In summary

$$\pi_{\phi,\theta}(\mathbf{x}^{(t)}, D^{(c)}) = \texttt{dec}_\theta(\mathbf{x}^{(t)}, r), \quad r = \texttt{enc}_\phi(D^{(c)}). \tag{1}$$

In CNPs, a standard choice, which we also use here, is to let $\pi_{\phi,\theta}(\mathbf{x}^{(t)}, D^{(c)})$ return a mean $\mu_{\phi,\theta}(x^{(t)}, D^{(c)})$ and a variance $\sigma^2_{\phi,\theta}(x^{(t)}, D^{(c)})$, to parameterise a predictive distribution that factorises across the target points $y^{(t)}|x^{(t)} \sim \mathcal{N}(\mu_{\phi,\theta}(x^{(t)}, D^{(c)}), \sigma^2_{\phi,\theta}(x^{(t)}, D^{(c)}))$. We note that our framework straightforwardly extends to more complicated $\pi_{\phi,\theta}(\mathbf{x}^{(t)}, D^{(c)})$. To train a CNP to make accurate predictions, we can optimise a log-likelihood objective Garnelo et al. [2018a] such as

$$\mathcal{L}(\theta, \phi) = \mathbb{E}_D \left[ \log \mathcal{N} \left( \mathbf{y}^{(t)}_m | \mu_{\phi,\theta}(\mathbf{x}^{(t)}_m, D^{(c)}), \sigma^2_{\phi,\theta}(\mathbf{x}^{(t)}, D^{(c)}) \right) \right], \tag{2}$$

where the expectation is taken over the distribution over tasks $D$. This objective is optimised by presenting each task $D_m$ to the CNP, computing the gradient of the loss with back-propagation, and updating the parameters $(\phi, \theta)$ of the CNP with any first-order optimiser (see alg. 1). This process trains the CNP to make well-calibrated predictions for $D^{(t)}$ given $D^{(c)}$. At test time, given a new $D^{(c)}$, we can use $\pi_{\phi,\theta}$ which can be queried at arbitrary target inputs, to obtain corresponding predictions (alg. 2).

**Convolutional CNPs.** Whenever we have useful inductive biases or other prior knowledge, we can leverage these by building them directly into the encoder and the decoder of the CNP. Stationarity is a powerful inductive bias that is often encountered in potentially sensitive applications such as time series or spatio-temporal regression. Whenever the generating process is stationary, the corresponding Bayesian predictive posterior is TE [Foong et al., 2020]. ConvCNPs leverage this inductive bias using TE architectures [Cohen and Welling, 2016, Huang et al., 2023].

**ConvCNP encoder.** To achieve TE, the ConvCNP encoder produces an $r$ that is itself a TE function.

---

**Algorithm 1** Meta-training a neural process.

**Input:** Simulated datasets $(D_m)_{m=1}^M$, encoder $\texttt{enc}_\phi$, decoder $\texttt{dec}_\theta$, iterations $T$, optimiser $\texttt{opt}$
**Output:** Optimised parameters $\phi, \theta$
**for** $i \in \{1 \dots T\}$ **do**
    Choose $D$ from $(D_m)_{m=1}^M$ randomly
    $D^{(c)}, D^{(t)} \leftarrow D$
    $\mathbf{x}^{(t)}, \mathbf{y}^{(t)} \leftarrow D^{(t)}$
    $\boldsymbol{\mu}, \boldsymbol{\sigma}^2 \leftarrow \texttt{dec}_\theta(\mathbf{x}^{(t)}, \texttt{enc}_\phi(D^{(c)}))$
    $\mathcal{L}(\theta, \phi) \leftarrow \log \mathcal{N}(\mathbf{y}^{(t)} | \boldsymbol{\mu}, \boldsymbol{\sigma}^2)$
    $\phi, \theta \leftarrow \texttt{opt}(\phi, \theta, \nabla_{\phi,\theta}\mathcal{L})$
**end for**
**Return** $\phi, \theta$

---

**Algorithm 2** Meta-testing a neural process.

**Input:** Real context $D^{(c)}$, $\texttt{enc}_\phi$, $\texttt{dec}_\theta$
**Output:** Predictive $\mu, \sigma$, with domain $\mathcal{X}$
$\mu(\cdot), \sigma(\cdot) \leftarrow \texttt{dec}_\theta(\cdot, \texttt{enc}_\phi(D^{(c)}))$
**Return** $\mu, \sigma$

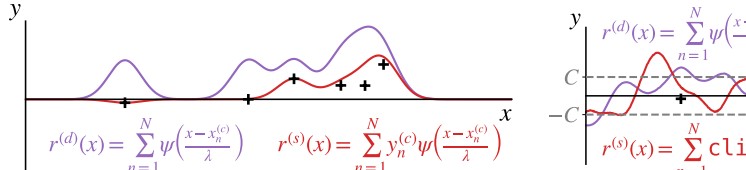

Figure 3: Left; Illustration of the ConvCNP encoder $\mathrm{enc}_\phi$. Black crosses show an example context set $D^{(c)}$. The density channel $r^{(d)}$ is shown in purple and the signal channel $r^{(r)}$ is shown in red. The representation $r$ consists of concatenating $r^{(d)}$ and $r^{(s)}$. Right; Illustration of the DPConvCNP encoder. Black crosses show an example context $D^{(c)}$, clipped with a threshold $C$ (gray dashed). Here, a single point (rightmost) is clipped (gray cross shows value before clipping). The density and signal channels are computed and GP noise is added to obtain the DP representation (red & purple).

Specifically, $\mathrm{enc}_\phi$ maps the context $D^{(c)} = ((x_n^{(c)}, y_n^{(c)}))_{n=1}^N$ to the function $r : \mathcal{X} \to \mathbb{R}^2$

$$
r(x) = \begin{bmatrix} r^{(d)}(x) \\ r^{(s)}(x) \end{bmatrix} = \sum_{n=1}^N \begin{bmatrix} 1 \\ y_n^{(c)} \end{bmatrix} \psi\left(\frac{x - x_n^{(c)}}{\lambda}\right), \tag{3}
$$

where $\psi$ is the Gaussian radial basis function (RBF) and $\phi = \{\lambda\}$. We refer to the two channels of $r$ as the *density* $r^{(d)}$ and the *signal* $r^{(s)}$ channels, which can be viewed as a smoothed version of $D^{(c)}$. The density channel carries information about the inputs of the context data, while the signal channel carries information about the outputs. This encoder is referred to as the SetConv.

**ConvCNP decoder.** Once $r$ has been computed, it is passed to the decoder which performs three steps. First, it discretises $r$ using a pre-specified resolution. Then, it applies a CNN to the discretised signal, and finally it uses an RBF smoother akin to Equation (3) to make predictions at arbitrary target locations. The aforementioned steps are all TE so, composing them with the TE encoder produces a TE prediction map [Bronstein et al., 2021]. The ConvCNP has universal approximator properties and produces state-of-the-art, well-calibrated predictions [Gordon et al., 2020].

## 3.2 Differential Privacy

Differential privacy [Dwork et al., 2006, Dwork and Roth, 2014] quantifies the maximal privacy loss to data subjects that can occur when the results of analysis are released. The loss is quantified by two numbers, $\epsilon$ and $\delta$, which bound the change in the distribution of the output of an algorithm, when the data of a single data subject in the dataset change.

**Definition 3.1.** *An algorithm $\mathcal{M}$ is $(\epsilon, \delta)$-DP if for neighbouring $D, D'$ and all measurable sets $S$*

$$
\Pr(\mathcal{M}(D) \in S) \leq e^\epsilon \Pr(\mathcal{M}(D') \in S) + \delta. \tag{4}
$$

We consider $D \in \mathbb{R}^{N \times d}$ with $N$ users and $d$ dimensions, and use the *substitution neighbourhood* relation $\sim_S$ where $D \sim_S D'$ if $D$ and $D'$ differ by at most one row.

**Gaussian DP.** In Section 3.3 we discuss the functional mechanism of Hall et al. [2013], which we use in the ConvCNP. However, the original privacy guarantees derived by Hall et al. [2013] are suboptimal. We improve upon these using the framework of Gaussian DP [GDP; Dong et al., 2022]. Dong et al. [2022] define GDP from a hypothesis testing perspective, which is not necessary for our purposes. Instead, we present GDP through the following conversion formula between GDP and DP.

**Definition 3.2.** *A mechanism $\mathcal{M}$ is $\mu$-GDP if and only if it is $(\epsilon, \delta(\epsilon))$-DP for all $\epsilon \geq 0$, where*

$$
\delta(\epsilon) = \Phi\left(-\frac{\epsilon}{\mu} + \frac{\mu}{2}\right) - e^\epsilon \Phi\left(-\frac{\epsilon}{\mu} - \frac{\mu}{2}\right) \tag{5}
$$

*and $\Phi$ is the CDF of the standard Gaussian distribution.*

**Properties of (G)DP.** Differential privacy has several useful properties. First, *post-processing immunity* guarantees that post-processing the result of a DP algorithm does not cause privacy loss:

**Theorem 3.3** (Dwork and Roth 2014). *Let $\mathcal{M}$ be an $(\epsilon, \delta)$-DP (or $\mu$-GDP) algorithm and let $f$ be any, possibly randomised, function. Then $f \circ \mathcal{M}$ is $(\epsilon, \delta)$-DP (or $\mu$-GDP).*

*Composition* of DP mechanisms refers to running multiple mechanisms on the same data. When each mechanism can depend on the outputs of the previous mechanisms, the composition is called *adaptive*. GDP is particularly appealing because it has a simple and tight composition formula:

**Theorem 3.4** (Dong et al. 2022). *The adaptive composition of $T$ mechanisms that are $\mu_i$-GDP ($i = 1, \ldots, T$), is $\mu$-GDP with $\mu = \sqrt{\mu_1^2 + \cdots + \mu_T^2}$.*

**Gaussian mechanism.** One of the central mechanisms to guarantee DP, is the Gaussian mechanism. This releases the output of a function $f$ with added Gaussian noise

$$\mathcal{M}(D) = f(D) + \mathcal{N}(0, \sigma^2 I), \tag{6}$$

where the variance $\sigma^2$ depends on the $l_2$-sensitivity of $f$, defined as

$$\Delta = \sup_{D \sim D'} ||f(D) - f(D')||_2. \tag{7}$$

**Theorem 3.5** (Dong et al. 2022). *The Gaussian mechanism with variance $\sigma^2 = \Delta^2/\mu^2$ is $\mu$-GDP.*

## 3.3 The Functional Mechanism

Now we turn to the functional mechanism of Hall et al. [2013]. Given a dataset $D \in \mathbb{R}^{N \times d}$, the functional mechanism releases a function $f_D \colon T \to \mathbb{R}$, where $T \subset \mathbb{R}^d$, with added noise from a Gaussian process. For simplicity, here we only define the functional mechanism for functions in a *reproducible kernel Hilbert space* (RKHS), and defer the more general definition to Appendix A.2.

**Definition 3.6.** *Let $g$ be a sample path of a Gaussian process having mean zero and covariance function $k$, and let $\mathcal{H}$ be an RKHS with kernel $k$. Let $\{f_D : D \in \mathcal{D}\} \subset \mathcal{H}$ be a family of functions indexed by datasets, satisfying*

$$\Delta_{\mathcal{H}} f \overset{\text{def}}{=} \sup_{D \sim D'} ||f_D - f_{D'}||_{\mathcal{H}} \leq \Delta. \tag{8}$$

*The functional mechanism with multiplier $c$ and sensitivity $\Delta$ is defined as*

$$\mathcal{M}(D) = f_D + cg. \tag{9}$$

**Theorem 3.7** (Hall et al.). *If $\epsilon \leq 1$, the mechanism in Def. 3.6 with $c = \frac{\Delta}{\epsilon}\sqrt{2\ln(2/\delta)}$ is $(\epsilon, \delta)$-DP.*

# 4 Differential privacy for the ConvCNP

Now we turn to our main contributions. First, we tighten the functional mechanism privacy analysis in Section 4.1 and then we build the functional mechanism into the ConvCNP in Section 4.2.

## 4.1 Improving the Functional Mechanism

The privacy bounds given by Theorem 3.7 are suboptimal, and do not allow us to use the tight composition formula from Theorem 3.4. However, the proof of Theorem 3.7 builds on the classical Gaussian mechanism privacy bounds, which we can replace with the GDP theory from Section 3.2. As demonstrated in Figure 4, our bound offers significantly smaller $\epsilon$ for the same noise standard deviation, compared to the existing bounds of Hall et al. [2013] and Jiang et al. [2023].

**Theorem 4.1.** *The functional mechanism with sensitivity $\Delta$ and multiplier $c = \Delta/\mu$ is $\mu$-GDP.*

*Proof.* The proof of Theorem 3.7 from Hall et al. [2013] shows that any $(\epsilon, \delta)$-DP bound for the Gaussian mechanism carries over to the functional mechanism. Replacing the classical Gaussian mechanism bound with the GDP bound proves the claim. For details, see Appendix A. $\square$

**Algorithm 3** DPSetConv; modifications to the original SetConv layer shown in blue.

---

**Input:** Grid $\mathbf{x} \subseteq \mathbb{R}^D$, $D^{(c)}$, $(\epsilon, \delta)$, RBF covariance $k$ with scale $\lambda$, threshold $C$, DP accounting method `noise_scales`.
**Output:** DP representation of $r^{(d)}, r^{(s)}$.
$\tilde{y}_n^{(c)} \leftarrow \texttt{clip}(y_n^{(c)}, C)$ for $n = 1, \ldots, N$
$g_d, g_s \sim \mathcal{GP}(0, k)$
$\mathbf{g}_d, \mathbf{g}_s \leftarrow g_d(\mathbf{x}), g_s(\mathbf{x})$
$\sigma_d, \sigma_s \leftarrow \texttt{noise\_scales}(\epsilon, \delta, C)$
$\mathbf{r}^{(d)} \leftarrow \sum_{n=1}^{N} \psi((\mathbf{x} - x_n^{(c)})/\lambda) + \sigma_d \mathbf{g}_d$
$\mathbf{r}^{(s)} \leftarrow \sum_{n=1}^{N} \tilde{y}_n^{(c)} \psi((\mathbf{x} - x_n^{(c)})/\lambda) + \sigma_s \mathbf{g}_s$
**Return:** Density and signal $\mathbf{r}^{(d)}, \mathbf{r}^{(s)}$.

---

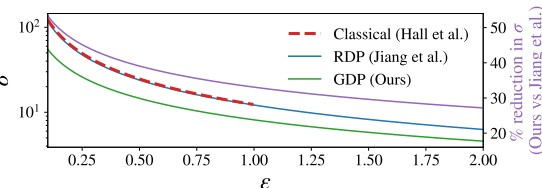

Figure 4: Noise magnitude comparison for the classical functional mechanism of Hall et al. [2013], the RDP-based mechanism of Jiang et al. [2023] and our improved GDP-based mechanism. The line for Hall et al. cuts off at $\epsilon = 1$ since their bound has only been proven for $\epsilon \leq 1$. We set $\Delta^2 = 10$ and $\delta = 10^{-3}$, which are representative values from our experiments. See Appendix A.6 for more details.

## 4.2 Differentially Private Convolutional CNP

**Differentially Private SetConv.** Now we turn to building the functional DP mechanism into the ConvCNP. We want to modify the SetConv encoder (Eq. 3) to make it DP. As a reminder, the SetConv outputs the density $r^{(d)}$ and signal $r^{(s)}$ channels

$$\begin{bmatrix} r^{(d)}(x) \\ r^{(s)}(x) \end{bmatrix} = \sum_{n=1}^{N} \begin{bmatrix} 1 \\ y_n^{(c)} \end{bmatrix} \psi\left(\frac{x - x_n^{(c)}}{\lambda}\right), \tag{10}$$

which are the two quantities we want to release under DP. To achieve this, we must first determine the sensitivity of $r^{(d)}$ and $r^{(s)}$, as defined in Eq. 8. Recall that we use the substitution neighbourhood relation $\sim_S$, defined as $D_1^{(c)} \sim_S D_2^{(c)}$ if $D_1^{(c)}$ and $D_2^{(c)}$ differ in at most one row, i.e. by a single context point. Since the RBF $\psi$ is bounded above by 1, it can be shown (see Appendix A.4) that the squared $l_2$-sensitivity of $r^{(d)}$ is bounded above by 2, and this bound is tight. Unfortunately however, since the signal channel $r^{(s)}$ depends linearly on each $y_n^{(c)}$ (see Eq. 10), its sensitivity is unbounded. To address this, we clip each $y_n^{(c)}$ by a threshold $C$, which is a standard way to ensure the sensitivity is bounded. With this modification we obtain the following tight sensitivities for $r^{(d)}$ and $r^{(s)}$:

$$\Delta_{\mathcal{H}}^2 r^{(d)} = 2, \quad \Delta_{\mathcal{H}}^2 r^{(s)} = 4C^2 \tag{11}$$

With these in place, we can state our privacy guarantee which forms the basis of the DPConvCNP. Post-processing immunity (Theorem 3.3) ensures that post-processing $r^{(s)}$ and $r^{(d)}$ with the ConvCNP decoder does not result in further privacy loss.

**Theorem 4.2.** *Let $g_d$ and $g_s$ be sample paths of two independent Gaussian processes having zero mean and covariance function $k$, such that $0 \leq k \leq C_k$ for some $C_k > 0$. Let $\Delta_d^2 = 2C_k$ and $\Delta_s^2 = 4C^2 C_k$. Then releasing $r^{(d)} + \sigma_d g_d$ and $r^{(s)} + \sigma_s g_s$ is $\mu$-GDP with $\mu = \sqrt{\Delta_s^2/\sigma_s^2 + \Delta_d^2/\sigma_d^2}$.*

*Proof.* The result follows by starting from the GDP bound of the mechanism in Theorem 4.1 and using Theorem 3.4 to combine the privacy costs for the releases of $r^{(d)}$ and $r^{(s)}$. □

**Corollary 4.3.** *Algorithm 2 with the DPSetConv encoder from Algorithm 3 is $(\epsilon, \delta)$-DP with respect to the real context set $D^{(c)}$.*

*Proof.* The `noise_scales` method in Algorithm 3 computes the appropriate $\sigma_d$ and $\sigma_s$ values from Theorem 4.2 and Definition 3.2 such that releasing the functional encodings $r^{(d)} + \sigma_d g_d$ and $r^{(s)} + \sigma_s g_s$ is $(\epsilon, \delta)$-DP. The $(\epsilon, \delta)$-DP guarantee extends [Hall et al., 2013, Proposition 5] to the point evaluations $\mathbf{r}^{(d)}$ and $\mathbf{r}^{(s)}$ over the grid $\mathbf{x}$ in Algorithm 3. Post-processing immunity (Theorem 3.3) extends $(\epsilon, \delta)$-DP to Algorithm 2. □

## 4.3 Training the DPConvCNP

**Training loss and algorithm.** We meta-train the DPConvCNP parameters $\theta, \phi$ using the CNP log-likelihood (eq. 2) within Algorithm 1, and meta-test it using alg. 2. Importantly, the encoder

$\text{enc}_\phi$ now includes clipping and adding noise (alg. 3) in its forward pass. Meta-training with the functional in place is crucial, because it teaches the decoder to handle the DP noise and clipping.

**Privacy hyperparameters.** By Definition 3.2 and Theorem 4.2, each $(\epsilon, \delta)$-budget implies a $\mu$-budget, placing a constraint on the sensitivities and noise magnitudes, namely $\mu^2 = \Delta_s^2/\sigma_s^2 + \Delta_d^2/\sigma_d^2$. Since $\psi$ is an RBF, $\Delta_d^2 = 2$ and $\Delta_s^2 = 4C^2$, and we need to specify $C, \sigma_s$ and $\sigma_d$, subject to this constraint. We introduce a variable $0 < t < 1$ and rewrite the constraint as

$$\sigma_s^2 = \frac{4C^2}{t\mu^2} \quad \text{and} \quad \sigma_d^2 = \frac{2}{(1-t)\mu^2} \tag{12}$$

allowing us to freely set $t$ and $C$. One straightforward approach is to fix $t$ and $C$ to hand-picked values, but this is sub-optimal since the optimal values depend on $\mu, N$, and the data statistics. Instead, we can make them adaptive, letting $t : \mathbb{R}^+ \times \mathbb{N} \to (0, 1)$ and $C : \mathbb{R}^+ \times \mathbb{N} \to \mathbb{R}^+$ be learnable functions, e.g. neural networks $t(\mu, N) = \text{sig}(\text{NN}_t(\mu, N))$ and $C(\mu, N) = \exp(\text{NN}_C(\mu, N))$ where $\text{sig}$ is the sigmoid. These networks are meta-trained along with all other parameters of the DPConvCNP.

## 5 Experiments & Discussion

We conduct experiments on synthetic and a sim-to-real task with real data. We provide the exact experimental details in Appendix E. We make our implementation of the DPConvCNP public in the repository `https://github.com/cambridge-mlg/dpconvcnp`.

**DP-SGD baseline.** Since, we are interested in the small-data regime, i.e. a few hundred datapoints per task, we turn to Gaussian processes [GP; Rasmussen and Williams, 2006], the gold-standard model for well-calibrated predictions in this setting. To enforce DP, we make the GP variational [Titsias, 2009], and use DP-SGD [Abadi et al., 2016] to optimise its variational parameters and hyperparameters. This is a strong baseline because GPs excel in small data, and DP-SGD is a state-of-the-art DP fine-tuning algorithm. We found it critical to carefully tune the DP-SGD parameters and the GP initialisation using BayesOpt, and devoted substantial compute on this to ensure we have maximised GP performance. We refer to this baseline as the DP-SVGP. For details see Appendix D.

**General setup.** In both synthetic and sim-to-real experiments, we first tuned the DP as well as the GP initialisation parameters of the DP-SVGP on synthetic data using BayesOpt. We then trained the DPConvCNP on synthetic data from the same generative process. Last, we tested both models on unseen test data. For the DP-SVGP, testing involves DP fine-tuning its variational parameters and its hyperparameters on each test set. For the DPConvCNP, testing involves a single forward pass through the network. We report results in Figures 6 and 7, and discuss them below.

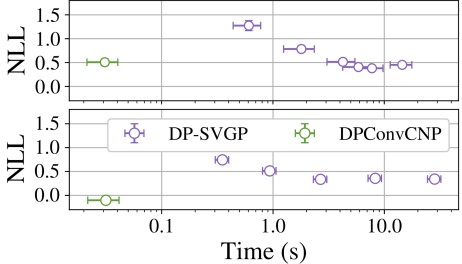

Figure 5: Deployment-time comparison on Gaussian (top) and non-Gaussian (bottom) data. We ran the DP-SVGP for different numbers of DP-SGD steps to determine a speed versus quality-of-fit tradeoff. Reporting 95% confidence intervals.

### 5.1 Synthetic tasks

**Gaussian data.** First, we generated data from a GP with an exponentiated quadratic (EQ) covariance (Figure 6; top), fixing its signal and noise scales, as well as its lengthscale $\ell$. For each $\ell$ we sampled datasets with $N \sim \mathcal{U}[1, 512]$ and privacy budgets with $\epsilon \sim \mathcal{U}[0.90, 4.00]$ and $\delta = 10^{-3}$. We trained separate DP-SVGPs and DPConvCNPs for each $\ell$ and tested them on unseen data from the same generative process (*non-amortised*; Figure 6). These models can handle different privacy budgets but only work well for the lengthscale they were trained on. In practice an appropriate lengthscale is not known *a priori*. To make this task more realistic, we also trained a single DPConvCNP on data with randomly sampled $\ell \sim \mathcal{U}[0.25, 2.00]$ (*amortised*; Figure 6). This model implicitly infers $\ell$ and simultaneously makes predictions, under DP. We also show the performance of the non-DP Bayes posterior, which is optimal (*oracle*; Figure 6 top). See Appendix E.1 for more details.

**DPConvCNP competes with DP-SVGP.** Even in the Gaussian setting, where the DP-SVGP is given the covariance of the generative process, the DPConvCNP remains competitive (red and

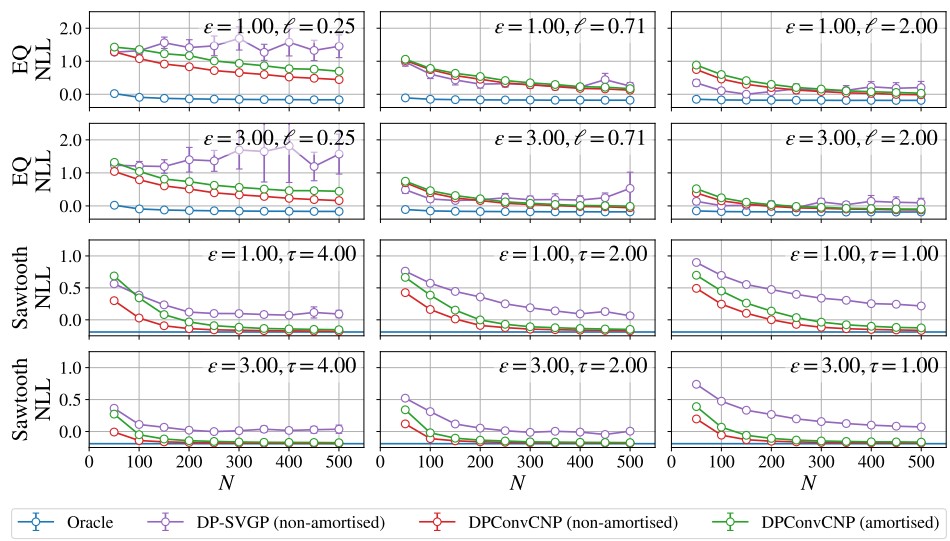

Figure 6: Negative log-likelihoods (NLL) of the DPConvCNP and the DP-SVGP baseline on synthetic data from a EQ GP (top two rows; EQ lengthscale $\ell$) and non-Gaussian data from sawtooth waveforms (bottom two rows; waveform period $\tau$). For each point shown we report the mean NLL with its 95% confidence intervals (error bars too small to see). See Appendix C.2 for example fits.

purple in Figure 6; top). While the DP-SVGP outperforms the DPConvCNP for some $N$ and $\ell$, the gaps are typically small. In contrast, the DP-SVGP often fails to provide sensible predictions (see $\ell = 0.25, N \geq 300$), and tends to overestimate the lengthscale, which is a known challenge in variational GPs [Bauer et al., 2016]. We also found that the DP-SVGP tends to underestimate the observation noise, resulting in over-smoothed *and* over-confident predictions which lead to a counter-intuitive reduction in performance as $N$ increases. By contrast, the DPConvCNP gracefully handles different $N$ and recovers predictions that are close to the non-DP Bayesian posterior for modest $\epsilon$ and $N$, with runtimes several orders of magnitude faster than the DP-SVGP (Figure 5).

**Amortising over $\ell$ and privacy budgets.** We observe that the DPConvCNP trained on a range of lengthscales (green; Figure 6) accurately infers the lengthscale of the test data, with only a modest performance reduction compared to its non-amortised counterpart (red). The ability of the DPConvCNP to implicitly infer $\ell$ while making calibrated predictions is remarkable, given the DP constraints under which it operates. Further, we observe that the DPConvCNP works well across a range of privacy budgets. In preliminary experiments, we found that the performance loss due to amortising over privacy budgets is small. This is particularly appealing because a single DPConvCNP can be trained on a range of budgets and deployed at test time using the privacy level specified by the practitioner, eliminating the need for separate models for different budgets.

**Non-Gaussian synthetic tasks.** We generated data from a non-Gaussian process with sawtooth signals, which has previously been identified as a challenging task Bruinsma et al. [2023]. We sampled the waveform direction and phase using a fixed period $\tau$ and adding Gaussian observation noise with a fixed magnitude. We gave the DP-SVGP an advantage by using a periodic covariance function, and truncating the Fourier series of the waveform signal to make it continuous: otherwise, since the DP-SVGP cannot handle discontinuities in the sawtooth signal, it explains the data mostly as noise, failing catastrophically. Again, we trained a separate DP-SVGP and DPConvCNP for each $\tau$, as well as a single DPConvCNP model on randomly sampled $\tau^{-1} \sim \mathcal{U}[0.20, 1.25]$. We report results in Figure 6 (bottom), along with a non-DP oracle (blue). The Bayes posterior is intractable, so we report the average NLL of the observation noise, which is a lower bound to the NLL.

**DPConvCNP outperforms the DP-SVGP.** We find that, even though we gave the DP-SVGP significant advantages, the DPConvCNP still outperforms it, and produces near-optimal predictions even for modest $N$ and $\epsilon$. Overall, our findings in the non-Gaussian tasks mirror those of the Gaussian tasks. The DPConvCNP can amortise over different signal periods with very small performance drops (red, green in Figure 6; bottom). Given the difficulty of this task, the fact that the DPConvCNP can predict accurately for signals with different periods under DP constraints is especially impressive.

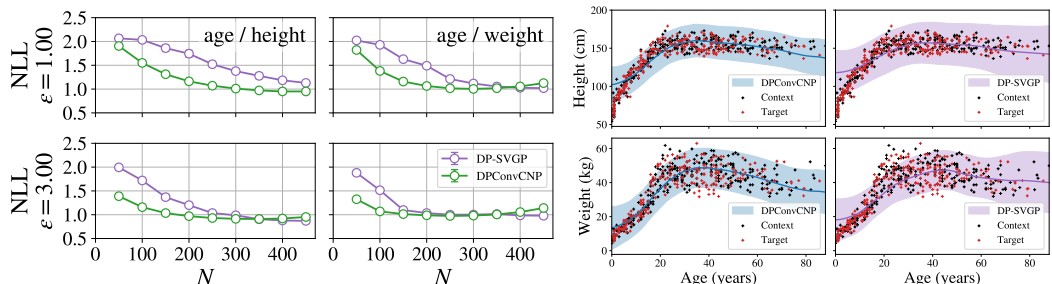

Figure 7: Left; Negative log-likelihoods of the DPConvCNP and the DP-SVGP baseline on the sim to real task with the !Kung dataset, predicting individuals' height from their age (left col.) or their weight from their age (right col.). For each point shown here, we partition each dataset into a context and target at random, make predictions, and repeat this procedure 512 times. We report mean NLL with its 95% confidence intervals. Error bars are to small to see here. Right; Example predictions for the DPConvCNP and the DP-SVGP, showing the mean and 95% confidence intervals, with $N = 300, \epsilon = 1.00, \delta = 10^{-3}$. The DPConvCNP is visibly better-calibrated than the DP-SVGP.

## 5.2 Sim-to-real tasks

**Sim-to-real task.** We evaluated the performance of the DPConvCNP in a sim-to-real task, where we train the model on simulated data and test it on the the Dobe !Kung dataset [Howell, 2009], also used by Smith et al. [2018], containing age, weight and height measurements of 544 individuals. We generated data from GPs with a Matérn-$3/2$ covariance, with a fixed signal scale of $\sigma_v = 1.00$, randomly sampled noise scale $\sigma_n \sim \mathcal{U}[0.20, 0.60]$ and lengthscale $\ell \sim \mathcal{U}[0.50, 2.00]$. We chose Matérn-$3/2$ since its paths are rougher than those of the EQ, and picked hyperparameter ranges via a back-of-the envelope calculation, without tuning them for the task. We trained a single DP-SVGP and a DPConvCNP with $\epsilon \sim \mathcal{U}[0.90, 4.00]$ and $\delta = 10^{-3}$. We consider two test tasks: predicting the height or the weight of an individual from their age. For each $N$, we split the dataset into a context and target at random, repeating the procedure for multiple splits.

**Sim-to-real comparison.** While the two models perform similarly for large $N$, the DPConvCNP performs much better for smaller $N$ (Figure 7; left). The DPConvCNP predictions are surprisingly good even for strong privacy guarantees, e.g. $\epsilon = 1.00, \delta = 10^{-3}$, and a modest dataset size (Figure 7; right), and significantly better-calibrated than those of the DP-SVGP, which under-fits. Note we have not tried to tune the simulator or add prior knowledge, which could further improve performance.

## 6 Limitations & Conclusion

**Limitations.** The DPConvCNP does not model dependencies between target outputs, which is a major limitation. This could be achieved straightforwardly by extending our approach to LNPs, GNPs, or ARNPs. Another limitation is that the efficacy of any sim-to-real scheme is limited by the quality of the simulated data. If the real and the simulated data differ substantially, then sim-to-real transfer has little hope of working. This can be mitigated by simulating diverse datasets to ensure the real data are in the training distribution. However, as simulator diversity increases, predictions typically become less certain, so there is a sweet spot in simulator diversity. While we observed strong sim-to-real results, exploring the effect of this diversity is a valuable direction for future work.

**Broader Impacts.** This paper presents work whose goal is to advance the field of DP. Generally, we view the potential for broader impact of this work as generally positive. Ensuring individual user privacy is critical across a host of Machine Learning applications. We believe that methods such as ours, aimed at improving the performance of DP algorithms and improve their practicality, have the potential to have a positive impact on individual users of Machine Learning models.

**Conclusion.** We proposed an approach for DP meta-learning using NPs. We leveraged and improved upon the functional DP mechanism of Hall et al. [2013], and showed how it can be naturally built into the ConvCNP to protect the privacy of the meta-test set with DP guarantees. Our improved bounds for the functional DP mechanism are substantial, providing the same privacy guarantees with a $\approx 30\%$ lower noise magnitude, and are likely of independent interest. We showed that the DPConvCNP is competitive and often outperforms a carefully tuned DP-SVGP baseline on both Gaussian and non-Gaussian synthetic tasks, while simultaneously being orders of magnitude faster at meta-test time. Lastly, we demonstrated how the DPConvCNP can be used as a sim-to-real model in a realistic evaluation scenario in the small data regime, where it outperforms the DP-SVGP baseline.

## Acknowledgements

This work was supported in part by the Research Council of Finland (Flagship programme: Finnish Center for Artificial Intelligence, FCAI as well as Grants 356499 and 359111), the Strategic Research Council at the Research Council of Finland (Grant 358247) as well as the European Union (Project 101070617). Views and opinions expressed are however those of the author(s) only and do not necessarily reflect those of the European Union or the European Commission. Neither the European Union nor the granting authority can be held responsible for them. SM is supported by the Vice Chancellor's and Marie and George Vergottis Scholarship, and the Qualcomm Innovation Fellowship. Richard E. Turner is supported by Google, Amazon, ARM, Improbable, EPSRC grant EP/T005386/1, and the EPSRC Probabilistic AI Hub (ProbAI, EP/Y028783/1).

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

# A  Differential Privacy Details

## A.1  Measure-Theoretic Details

Definition 3.1 is the typical definition of $(\epsilon, \delta)$-DP that is given in the literature, but it glosses over some measure-theoretic details that are usually not important, but are important for the functional mechanism. In particular, the precise meaning of "measurable" is left open. Here, we make the $\sigma$-field that "measurable" implicitly refers to explicit:

**Definition A.1.** *An algorithm $\mathcal{M}$ is $(\epsilon, \delta, \mathcal{A})$-DP for a $\sigma$-field $\mathcal{A}$ if, for neighbouring datasets $D, D'$ and all $A \in \mathcal{A}$,*

$$\Pr(\mathcal{M}(D) \in A) \le e^\epsilon \Pr(\mathcal{M}(D') \in A) + \delta. \tag{13}$$

Hall et al. [2013] point out that the choice of $\mathcal{A}$ is important, and insist that $\mathcal{A}$ be the finest $\sigma$-field on which $\mathcal{M}(D)$ is defined for all $D$. When the output of the mechanism is discrete, or an element of $\mathbb{R}^n$, this corresponds with the $\sigma$-field that is typically implicitly used in such settings. When the output is a function, as in the functional mechanism, the choice of $\mathcal{A}$ is not as clear [Hall et al., 2013]. Note that $\mathcal{A}$ is similarly implicitly present in the definition of GDP (Definition 3.2).

Next, we recall the construction of the appropriate $\sigma$-field for the functional mechanism from Hall et al. [2013]. Let $T$ be an index set. We denote the set of functions from $T$ to $\mathbb{R}$ as $\mathbb{R}^T$. For $S = (x_1, \ldots, x_n) \in T^n$ and a Borel set $B \in \mathcal{B}(\mathbb{R}^n)$,

$$C_{S,B} = \{f \in \mathbb{R}^T \mid (f(x_1), \ldots, f(x_n)) \in B\} \tag{14}$$

is called a cylinder set of functions. Let $\mathcal{C}_S = \{C_{S,B} \mid B \in \mathcal{B}(\mathbb{R}^n)\}$ and

$$\mathcal{F}_0 = \bigcup_{S : |S| < \infty} \mathcal{C}_S. \tag{15}$$

$\mathcal{F}_0$ is called the field of cylinder sets. $(\epsilon, \delta, \mathcal{F}_0)$-DP [2] amounts to $(\epsilon, \delta, \mathcal{B}(\mathbb{R}^n))$-DP for any evaluation of $f$ at a finite vector of points $(x_1, \ldots, x_n) \in T^n$, of any size $n \in \mathbb{N}$ [Hall et al., 2013].

The $\sigma$-field for the functional mechanism is the $\sigma$-field $\mathcal{F}$ generated by $\mathcal{F}_0$ [Hall et al., 2013]. It turns out that $(\epsilon, \delta, \mathcal{F}_0)$-DP is sufficient for $(\epsilon, \delta, \mathcal{F})$-DP.

## A.2  General Definition of the Functional Mechanism

**Definition A.2.** *Let $g$ be a sample path of a Gaussian process having mean zero and covariance function $k$. Let $\{f_D : D \in \mathcal{D}\} \subset \mathbb{R}^T$ be a family of functions indexed by datasets satisfying the inequality*

$$\sup_{D \sim D'} \sup_{n < \infty} \sup_{(x_1, \ldots, x_n) \in T^n} \left\| \Delta_{D,D'}^{(x_1, \ldots, x_n)} \right\|_2 \le \Delta, \tag{16}$$

*with*

$$\Delta_{D,D'}^{(x_1, \ldots, x_n)} = M^{-1/2}(x_1, \ldots, x_n) \begin{bmatrix} f_D(x_1) - f_{D'}(x_1) \\ \vdots \\ f_D(x_n) - f_{D'}(x_n) \end{bmatrix},$$

*where $M(x_1, \ldots x_n)_{ij} = k(x_i, x_j)$. The functional mechanism with multiplier $c$ and sensitivity $\Delta$ is defined as*

$$\mathcal{M}(D) = f_D + cg. \tag{17}$$

If $f$ is a member of a *reproducible kernel Hilbert space* (RKHS) $\mathcal{H}$ with the same kernel $k$ as the noise process $g$, the sensitivity bound of Definition A.2 is much simpler:

**Lemma A.3** (Hall et al. 2013). *For a function $f$ in an RKHS $\mathcal{H}$ with kernel $k$,*

$$\Delta_{\mathcal{H}} f \overset{\text{def}}{=} \sup_{D \sim D'} \|f_D - f_{D'}\|_{\mathcal{H}} \le \Delta. \tag{18}$$

*implies* (16).

---

[2]This is a small abuse of notation, as $\mathcal{F}_0$ is not a $\sigma$-field.

## A.3 Proof of Theorem 4.1

To prove Theorem 4.1, we need a GDP version of a lemma from Hall et al. [2013]:

**Lemma A.4.** *Suppose that, for a positive definite symmetric matrix $M \in \mathbb{R}^{d \times d}$, the function $f \colon \mathcal{D} \to \mathbb{R}^d$ satisfies*

$$\sup_{D \sim D'} ||M^{-1/2}(f(D) - f(D'))||_2 \leq \Delta. \tag{19}$$

*Then the mechanism $\mathcal{M}$ that outputs (Gaussian mechanism)*

$$\mathcal{M}(D) = f(D) + cZ, \quad Z \sim \mathcal{N}_d(0, M)$$

*is $\mu$-GDP with $c = \frac{\Delta}{\mu}$.*

*Proof.* We can write

$$\mathcal{M}(D) = M^{1/2}c\left(\frac{M^{-1/2}}{c}f(D) + S\right), \quad S \sim \mathcal{N}_d(0, I).$$

Denote $\mathcal{M}'(D) = \frac{M^{-1/2}}{c}f(D) + S$. $\mathcal{M}'$ is a Gaussian mechanism with variance 1. Because of (19), $\frac{M^{-1/2}}{c}f(D)$ has sensitivity

$$\Delta^* = \sup_{D \sim D'} \left\|\frac{M^{-1/2}}{c}f(D) - \frac{M^{-1/2}}{c}f(D')\right\|_2 \leq \frac{\Delta}{c}$$

so $\mathcal{M}'$ is $\mu$-GDP by Theorem 3.5. $\mathcal{M}$ is obtained by post-processing $\mathcal{M}'$, so it is also $\mu$-GDP. $\square$

**Theorem 4.1.** *The functional mechanism with sensitivity $\Delta$ and multiplier $c = \Delta/\mu$ is $\mu$-GDP.*

*Proof.* Let $T$ be the index set of the Gaussian process $G$, and let $S = (x_1, \ldots, x_n) \in T^n$. Then $(G(x_1), \ldots, G(x_n))$ has a multivariate Gaussian distribution with mean zero and covariance $\mathrm{Cov}(G(x_i), G(x_j)) = K(x_i, x_j)$. Then the vector obtained by evaluating $\mathcal{M}(D)$ at $(x_1, \ldots, x_n)$ is $\mu$-GDP by Lemma A.4, as (16) implies the sensitivity bound (19). Theorem 3.5 gives a curve of $(\epsilon, \delta(\epsilon))$-bounds for all $\epsilon \geq 0$ from $\mu$.

This holds for any $S \in T^n$ and any $n \in \mathbb{N}$, so $\mathcal{M}$ is $(\epsilon, \delta(\epsilon), \mathcal{F}_0)$-DP for all $\epsilon \geq 0$, which immediately implies $(\epsilon, \delta(\epsilon), \mathcal{F})$-DP. This curve can be converted back to $\mu$-GDP (with regards to $\mathcal{F}$) using Theorem 3.5. $\square$

## A.4 Functional Mechanism Sensitivities for DPConvCNP

To bound the sensitivity of $r^{(d)}$ and $r^{(s)}$ for the functional mechanism, we look at two neighbouring context sets $D_1^{(c)} = ((x_{n,1}^{(c)}, y_{n,1}^{(c)}))_{n=1}^N$ and $D_2^{(c)} = ((x_{n,2}^{(c)}, y_{n,2}^{(c)}))_{n=1}^N$ that differ only in the points $(x_1, y_1) \in D_1^{(c)}$ and $(x_2, y_2) \in D_2^{(c)}$. Let $r_{D_i^{(c)}}^{(d)}$ for $i \in \{1, 2\}$ denote $r^{(d)}$ from (10) computed from $D_i^{(c)}$, and define $r_{D_i^{(c)}}^{(s)}$ similarly.

Denote the RKHS of the kernel $k$ by $\mathcal{H}$. The distance in $\mathcal{H}$ between the functions $k_{x_1} = k(x_1, \cdot)$ and $k_{x_2} = k(x_2, \cdot)$ is given by

$$||k_{x_1} - k_{x_2}||_{\mathcal{H}}^2 = \langle k_{x_1} - k_{x_2}, k_{x_1} - k_{x_2}\rangle_{\mathcal{H}} \tag{20}$$
$$= k(x_1, x_1) - 2k(x_1, x_2) + k(x_2, x_2) \tag{21}$$
$$\leq 2C_k. \tag{22}$$

For the RBF kernel, this is a tight bound without other assumptions on $x$, as $k(x, x) = 1 = C_k$ for all $x$ and $k(x_1, x_2)$ can be made arbitrarily small by placing $x_1$ and $x_2$ far away from each other.

The sensitivity of $r^{(d)}$ for the functional mechanism can be bounded with (22): for ,

$$\Delta_{\mathcal{H}}^2 r^{(d)} = \sup_{D_1^{(c)} \sim_S D_2^{(c)}} \left\| r_{D_1^{(c)}}^{(d)} - r_{D_2^{(c)}}^{(d)} \right\|_{\mathcal{H}}^2 \tag{23}$$

$$= \sup_{D_1^{(c)} \sim_S D_2^{(c)}} \left\| \sum_{n=1}^N (k_{x_{n,1}^{(c)}} - k_{x_{n,2}^{(c)}}) \right\|_{\mathcal{H}}^2 \tag{24}$$

$$= \sup_{x_1, x_2} \| k_{x_1} - k_{x_2} \|_{\mathcal{H}}^2 \tag{25}$$

$$\leq 2C_k, \tag{26}$$

where the second to last line follows from the fact that $D_1^{(c)}$ and $D_2^{(c)}$ only differ in one datapoint. This is a tight bound for the RBF kernel, because when $x = x_1$, $k_{x_1}(x) = 1$ and $k_{x_2}(x) = 1$ can be a made arbitrarily small by moving $x_2$ far away from $x$.

For $r^{(s)}$ and any function $\phi$ with $|\phi(y)| \leq C$, we first bound

$$\| \phi(y_1) k_{x_1} - \phi(y_2) k_{x_2} \|_{\mathcal{H}}^2 \tag{27}$$

$$= \phi(y_1)^2 k(x_1, x_1) - 2\phi(y_1)\phi(y_2) k(x_1, x_2) + \phi(y_2)^2 k(x_2, x_2) \tag{28}$$

$$\leq 4C^2 C_k. \tag{29}$$

Again, these are tight bounds for the RBF kernel if we don't constrain $x$ or $y$ further.

The $\mathcal{H}$-sensitivity for $r^{(s)}$ is then derived in the same way as the sensitivity for $r^{(d)}$, giving

$$\Delta_{\mathcal{H}}^2 r^{(s)} \leq 4C^2 C_k. \tag{30}$$

### A.5 Gaussian Mechanism for DPConvCNP

A naive way of releasing $r(x)$ under DP is to first select discretisation points $x_1, \ldots, x_n$, in some way, and release $r(x_1), \ldots, r(x_n)$ with the Gaussian mechanism. The components of $r$, $r^{(s)}$ and $r^{(d)}$, have the following sensitivities:

$$\Delta^2 r^{(d)}(x) = \sup_{D_1^{(c)} \sim_S D_2^{(c)}} \left\| r_{D_1^{(c)}}^{(d)}(x) - r_{D_2^{(c)}}^{(d)}(x) \right\|_2^2 \tag{31}$$

$$= \sup_{D_1^{(c)} \sim_S D_2^{(c)}} \left| \sum_{n=1}^N (k_{x_{n,1}^{(c)}}(x) - k_{x_{n,2}^{(c)}}(x)) \right|^2 \tag{32}$$

$$= \sup_{x_1, x_2} | k_{x_1}(x) - k_{x_2}(x) |^2 \tag{33}$$

$$\leq C_k^2. \tag{34}$$

Line (33) follows from the fact that $D_1^{(c)}$ and $D_2^{(c)}$ only differ in one datapoint.

For $r^{(s)}(x)$, we have

$$| \phi(y_1) k_{x_1}(x) - \phi(y_2) k_{x_2}(x) |^2 \leq 4C^2 C_k^2. \tag{35}$$

Then we get

$$\Delta^2 r^{(s)}(x) \leq 4C^2 C_k^2 \tag{36}$$

following the derivation of $\Delta^2 r^{(d)}(x)$. For the RBF kernel, this is again tight without additional assumptions on $y$ or $x$.

These sensitivities give the following privacy bound:

**Theorem A.5.** *Let* $\Delta_s^2 = 4C^2 C_k^2$ *and* $\Delta_d^2 = C_k^2$. *Releasing* $n$ *evaluations of* $r(x) = (r^{(d)}(x), r^{(s)}(x))$ *with the Gaussian mechanism with noise variance* $\sigma^2$ *is* $\mu$-*GDP for*

$$\mu = \sqrt{n \frac{\Delta_s^2 + \Delta_d^2}{\sigma^2}}. \tag{37}$$

*Proof.* Releasing $n$ evaluations of $r(x)$ is simply an $n$-fold composition of Gaussian mechanisms that release $r(x)$ for one value. Releasing $r(x)$ for one value is a composition of releasing $r^{(d)}(x)$ and $r^{(s)}(x)$, which have the sensitivities $\Delta_s$ and $\Delta_d$. Now Theorems 3.5 and 3.4 prove the claim. $\quad\square$

As $\mu$ scales with $n$, this method must add a large amount of noise for even moderate numbers of discretisation points.

The difference between having a factor of $C_k^2$ in the $L_2$-sensitivities and $C_k$ in the $\mathcal{H}$-sensitivities is explained by the fact that the kernel also directly affects the noise variance for the functional mechanism, but it does not directly affect the noise variance with the Gaussian mechanism. This can be illustrated by considering what happens when the kernel is multiplied by a constant $u > 0$. This multiplies $C_k$ by $u$, and multiplies the $L_2$-sensitivities by $u^2$. For the Gaussian mechanism, this means multiplying the noise standard deviation by $u$, but simultaneously multiplying all released values by $u$, which does not change the signal-to-noise ratio. For the functional mechanism, multiplying the kernel values effectively multiplies $c$ by $\sqrt{u}$ and the squared sensitivities by $u$, which then cancel each other in $\mu$.

For the RBF kernel and clipping function $\phi$ with threshold $C = 1$, we see that $\Delta_{\mathcal{H}}^2 r^{(d)} = 2$ while $\Delta_2^2 r^{(d)} = 1$, and $\Delta_{\mathcal{H}}^2 r^{(s)} = 4$, while $2\Delta_2^2 r^{(s)} = 4$, so the functional mechanism adds noise with slightly more variance as releasing a single value with the Gaussian mechanism, so the functional mechanism adds noise of less variance when 2 or more discretisation points are required. However, the functional mechanism adds correlated noise, which is not as useful for denoising as the uncorrelated noise that the Gaussian mechanism adds.

### A.6 Details on Figure 4

In this section, we go over the details of the calculations behind Figure 4. The "classical" line of the figure is computed from Theorem 3.7. The GDP line uses Definition 3.2 to convert the $(\epsilon, \delta)$-pair into a GDP $\mu$ bound by numerically solving for $\mu$ in Eq.(5). $\sigma$ is then found with Theorem 4.1.

For the RDP line, we get an RDP guarantee from Corollary 2 of Jiang et al. [2023], which we convert to $(\epsilon, \delta)$ with Proposition 3 of Jiang et al. [2023]. These give the equation

$$\epsilon = \frac{\alpha \Delta^2}{2\sigma^2} - \frac{\ln \delta}{\alpha - 1}, \tag{38}$$

where $\alpha > 1$ is a parameter of RDP that can be freely chosen. The $\alpha$ value that minimises $\epsilon$ is

$$\alpha^* = \sqrt{-\frac{2\sigma^2 \ln \delta}{\Delta^2}} + 1. \tag{39}$$

Plugging $\alpha^*$ into Eq. (38) gives the quadratic equation

$$-\epsilon \sigma^2 + 2\sqrt{-\frac{\Delta^2 \ln \delta}{2}} \sigma + \frac{\Delta^2}{2} = 0 \tag{40}$$

that can be solved for $\sigma$.

To see that choosing the $\alpha$ that minimises $\epsilon$ also leads to the smallest $\sigma$ that satisfies a given $(\epsilon, \delta)$-bound, let

$$\epsilon(\alpha, \sigma) = \frac{\alpha \Delta^2}{2\sigma^2} - \frac{\ln \delta}{\alpha - 1}, \tag{41}$$

and let $\sigma^*$ be the solution to Eq. (40). Let $\alpha, \sigma$ be another pair that satisfies the $(\epsilon, \delta)$-bound. Since $\alpha^*$ is chosen to minimise $\epsilon$, $\epsilon(\alpha^*(\sigma), \sigma) \leq \epsilon(\alpha, \sigma)$. We can assume that $\epsilon(\alpha, \sigma) = \epsilon(\alpha^*(\sigma), \sigma) = \epsilon$, since otherwise we could reduce $\sigma$ further. Now

$$\epsilon(\alpha^*(\sigma^*), \sigma^*) = \epsilon = \epsilon(\alpha^*(\sigma), \sigma) \tag{42}$$

so $\epsilon(\alpha^*(\sigma^*), \sigma^*) = \epsilon(\alpha^*(\sigma), \sigma)$. By manipulating Eq. (40), we can see that $\epsilon(\alpha^*(\cdot), \cdot)$ is strictly decreasing, so this implies that $\sigma = \sigma^*$.

---

**Algorithm 4** Efficient sampling of GP noise on a $D$-dimensional grid.

---

**Input** : Dimension-wise grid location $u_d \in \mathbb{R}$, spacing $\gamma_d \in \mathbb{R}$ and number of points $N_d \in \mathbb{N}$,
  Product kernel $k : \mathbb{R}^D \times \mathbb{R}^D \to \mathbb{R}$ with factors $k_d : \mathbb{R} \times \mathbb{R} \to \mathbb{R}$.
**Output:** Sample $f_{n_1 \dots n_D}$ from GP with kernel $k$ on grid inputs $x_{n_1 \dots n_D}$ defined by $u_d, \gamma_d, N_d$.

Sample $f_{n_1 \dots n_D} \sim \mathcal{N}(0, 1)$ for each $1 \le n_d \le N_d, d = 1, \dots, D$ {Sample Gaussian noise}
**for** $d = 1$ **to** $D$ **do**
  $K_{dnm} \leftarrow k_d(u_d + n\gamma_d, u_d + m\gamma_d)$ for $0 \le n, m \le N_d - 1$ {Compute covariance}
  $L_d \leftarrow$ CHOLESKY($K_d$) {Compute Cholesky factor}
  $f \leftarrow$ MATMULALONGDIM($L_d, f, d$) {Matmul $f$ by $L_d$ along dimension $d$}
**end for**

---

## B  Efficient sampling of Gaussian process noise

In order to ensure differential privacy within the DPConvCNP, we need to add GP noise (from a GP with an EQ kernel) to the functional representation outputted by the SetConv. In practice, this is implemented by adding GP noise on the discretised representation, i.e. the data and density channels.

While sampling GP noise is typically tractable if the grid is one-dimensional, the computational and memory costs of sampling can easily become intractable for two- or three-dimensional grids. This is because the number of grid points increases exponentially with the number of input dimensions and, in addition to this, the cost of sampling increases cubically with the number of grid points. Fortunately, we can exploit the regularity of the grid and the fact that the EQ kernel is a product kernel, to make sampling tractable. Proposition B.1 illustrates how this can be achieved.

**Proposition B.1.** *Let $x \in \mathbb{R}^{N_1 \times \dots \times N_D}$ be a grid of points in $\mathbb{R}^D$ given by*

$$x_{n_1 \dots n_D} = (u_1 + (n_1 - 1)\gamma_1, \dots, u_D + (n_D - 1)\gamma_D), \tag{43}$$

*where $u_d \in \mathbb{R}$, $\gamma_d \in \mathbb{R}^+$ and $1 \le n_d \in \mathbb{N} \le N_d$ for each $d = 1, \dots, D$. Also let $k : \mathbb{R}^D \times \mathbb{R}^D \to \mathbb{R}$ be a product kernel, i.e. a kernel that satisfies*

$$k(z, z') = \prod_{d=1}^{D} k_d(z_d, z'_d), \tag{44}$$

*for some univariate kernels $k_d : \mathbb{R} \to \mathbb{R}$, for every $z, z' \in \mathbb{R}^D$, let*

$$K_{dnm} = k_d(u_d + (n - 1)\gamma_d, \ u_d + (m - 1)\gamma_d), \tag{45}$$

*and let $L_d$ be a Cholesky factor of the matrix $K_d$. Then if $\epsilon_{n_1 \dots n_D} \in \mathbb{R} \sim \mathcal{N}(0, 1)$ is a grid of corresponding i.i.d. standard Gaussian noise, the scalars $f_{n_1 \dots n_D} \in \mathbb{R}$, defined as*

$$f_{n_1 \dots n_D} = \sum_{k_1=1}^{N_1} L_{1 n_1 k_1} \cdots \sum_{k_D=1}^{N_D} L_{D n_D k_D} \ \epsilon_{k_1 \dots k_D}, \tag{46}$$

*are Gaussian-distributed, with zero mean and covariance*

$$\mathbb{C}[f_{n_1 \dots n_D}, f_{m_1 \dots m_D}] = k(x_{n_1 \dots n_D}, x_{m_1 \dots m_D}). \tag{47}$$

Before giving the proof of Proposition B.1, we provide pseudocode for this approach in Algorithm 4 and discuss the computation and memory costs of this implementation compared to a naive approach. Naive sampling on a grid of $N_1 \times \cdots \times N_D$ points requires computing a Cholesky factor for the covariance matrix of the entire grid and then multiplying standard Gaussian noisy by this factor. We discuss the costs of these operations, comparing them to the efficient approach.

**Computing Cholesky factors.** The cost of computing a Cholesky factor for covariance matrix of the entire $N_1 \times \cdots \times N_D$ grid incurs a computation cost of $\mathcal{O}(N_1^3 \times \cdots \times N_D^3)$ and a memory cost of $\mathcal{O}(N_1^2 \times \cdots \times N_D^2)$. By contrast, Algorithm 4 only ever computes Cholesky factors for $N_d \times N_d$ covariance matrices, so it incurs a computation cost of $\mathcal{O}\left(\sum_{d=1}^{D} N_d^3\right)$ and a memory cost of $\mathcal{O}\left(\sum_{d=1}^{D} N_d^2\right)$, which are both much lower than those of a naive implementation. For clarity, if

$N_1 = \cdots = N_D = N$, naive factorisation has $\mathcal{O}(N^{3D})$ computational and $\mathcal{O}(N^{2D})$ memory cost, whereas the efficient implementation has $\mathcal{O}(DN^3)$ computational and $\mathcal{O}(DN^2)$ memory cost.

**Matrix multiplications.** In addition, naively multiplying the Gaussian noise by the Cholesky factor of the entire covariance matrix incurs a $\mathcal{O}(N_1^2 \times \cdots \times N_D^2)$ computation cost. On the other hand, in Algorithm 4 we perform $D$ batched matrix-vector multiplications, where the $d^{\text{th}}$ multiplication consists of $\prod_{d'=d} N_{d'}$ matrix-vector multiplications, where a vector with $N_d$ entries is multiplied by an $N_d \times N_d$ matrix. The total computation cost for this step is only $\mathcal{O}\left(\sum_{d=1}^{D} N_d^2 \prod_{d' \neq d} N_{d'}\right)$. For example, if $N_1 = \cdots = N_D = N$, naive matrix-vector multiplication has a computation cost of $\mathcal{O}(N^{2D})$, whereas the efficient implementation has a computation cost of $\mathcal{O}(N^{D+1})$.

In Algorithm 4, CHOLESKY denotes a function that computes the Cholesky factor of a square positive-definite matrix. MATVECALONGDIM$(L_d, f, d)$ denotes the batched matrix-vector multiplication of an array $f$ by a matrix $L_d$ along dimension $d$, batching over the dimensions $d' \neq d$. Specifically, given a $D$-dimensional array $b$ with dimension sizes $N_1, \ldots, N_D$ and an $N_d \times N_d$ matrix $A$, the matrix-vector multiplication of $b$ by $A$ along dimension $d$ outputs the $D$-dimensional array

$$\text{MATVECALONGDIM}(A, b, d)_{n_1 \ldots n_D} = \sum_{j=1}^{N_d} A_{n_d j} b_{n_1 \ldots n_{d-1} \, j \, n_{d+1} \ldots n_D}. \tag{48}$$

From the above equation, we can see that initialising $f$ with standard Gaussian noise, and batch-multiplying $f$ by $L_d$ along dimension $d$ for each $d = 1, \ldots, D$, amounts to computing the nested sum in Equation (46). Note that the order with which these batch multiplications are performed does not matter: it does not change neither the numerical result nor the computation or memory cost of the algorithm.

*Proof of Proposition B.1.* From the definition above, we see that $f_{n_1 \ldots n_D}$ is a linear transformation of Gaussian random variables with zero mean, and therefore also has zero mean. For the covariance, again from the definition above, we have

$$\mathbb{C}\left[f_{n_1 \ldots n_D}, f_{m_1 \ldots m_D}\right] = \tag{49}$$

$$\mathbb{C}\left[\sum_{k_1=1}^{N_1} L_{1 n_1 k_1} \cdots \sum_{k_D=1}^{N_D} L_{D n_D k_D} \, \epsilon_{k_1 \ldots k_D}, \sum_{l_1=1}^{N_1} L_{1 m_1 l_1} \cdots \sum_{l_D=1}^{N_D} L_{D m_D l_D} \, \epsilon_{l_1 \ldots l_D}\right] = \tag{50}$$

$$\mathbb{C}\left[\sum_{k_1=1}^{N_1} \cdots \sum_{k_D=1}^{N_D} L_{1 n_1 k_1} \ldots L_{D n_D k_D} \, \epsilon_{k_1 \ldots k_D}, \sum_{l_1=1}^{N_1} \cdots \sum_{l_D=1}^{N_D} L_{1 m_1 l_1} \ldots L_{D m_D l_D} \, \epsilon_{l_1 \ldots l_D}\right] = \tag{51}$$

$$\mathbb{E}\left[\left(\sum_{k_1=1}^{N_1} \cdots \sum_{k_D=1}^{N_D} L_{1 n_1 k_1} \ldots L_{D n_D k_D} \, \epsilon_{k_1 \ldots k_D}\right)\left(\sum_{l_1=1}^{N_1} \cdots \sum_{l_D=1}^{N_D} L_{1 m_1 l_1} \ldots L_{D m_D l_D} \, \epsilon_{l_1 \ldots l_D}\right)\right], \tag{52}$$

where we have used the fact that the expectation of $f$ is zero. Now, expanding the product of sums above, taking the expectation and using the fact that $\epsilon_{n_1 \ldots n_D}$ are i.i.d., we see that all terms vanish, except those where $k_d = l_d$ for all $d = 1, \ldots, D$. Specifically, we have

$$\mathbb{C}\left[f_{n_1 \ldots n_D}, f_{m_1 \ldots m_D}\right] = \tag{53}$$

$$= \mathbb{E}\left[\sum_{k_1=1}^{N_1} \cdots \sum_{k_D=1}^{N_D} \sum_{l_1=1}^{N_1} \cdots \sum_{l_D=1}^{N_D} L_{1 n_1 k_1} \ldots L_{D n_D k_D} L_{1 m_1 l_1} \ldots L_{D m_D l_D} \, \epsilon_{k_1 \ldots k_D} \epsilon_{l_1 \ldots l_D}\right] \tag{54}$$

$$= \sum_{k_1=1}^{N_1} \cdots \sum_{k_D=1}^{N_D} \sum_{l_1=1}^{N_1} \cdots \sum_{l_D=1}^{N_D} L_{1 n_1 k_1} \ldots L_{D n_D k_D} L_{1 m_1 l_1} \ldots L_{D m_D l_D} \, \mathbb{E}\left[\epsilon_{k_1 \ldots k_D} \epsilon_{l_1 \ldots l_D}\right] \tag{55}$$

$$= \sum_{k_1=1}^{N_1} \cdots \sum_{k_D=1}^{N_D} \sum_{l_1=1}^{N_1} \cdots \sum_{l_D=1}^{N_D} L_{1 n_1 k_1} \ldots L_{D n_D k_D} L_{1 m_1 l_1} \ldots L_{D m_D l_D} \, \mathbb{1}_{k_1 = l_1, \ldots, \, k_D = l_D} \tag{56}$$

$$= \sum_{k_1=1}^{N_1} \cdots \sum_{k_D=1}^{N_D} L_{1 n_1 k_1} \ldots L_{D n_D k_D} L_{1 m_1 k_1} \ldots L_{D m_D k_D} \tag{57}$$

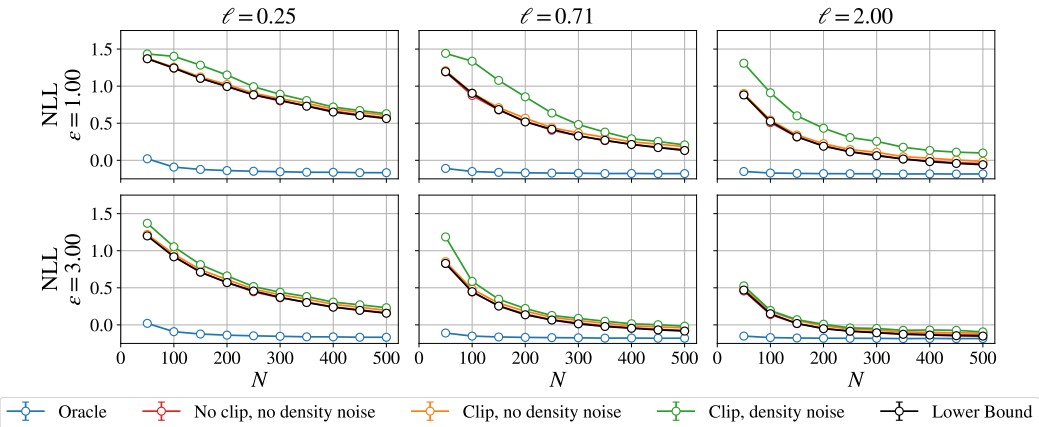

Figure S1: DPConvCNP performance on the GP modelling task, where the data are generated using an EQ GP with lengthscale $\ell$. We train three models per $\epsilon, \ell$ combination, keeping $\delta = 10^{-3}$ fixed as well as the clipping threshold $C = 2.00$ and noise weight $t = 0.50$ fixed. Specifically, we train one model where only noise to the signal channel (red; no clip, no density), one model where noise and clipping are applied to the signal channel (orange; clip, no density noise) and another model where noise and clipping to the signal channel as well as noise to the density channel are applied (green; clip, density noise). We also show the NLL of the oracle, non-DP, Bayesian posterior, which is the best average NLL that can be obtained on this task (blue). Lastly, we show a bound to the functional mechanism (black), which is a lower bound on the NLL that can be obtained with the functional mechanism with $C = 2.00, t = 0.50$ on this task. We used 512 evaluation tasks for each $N, \ell, \epsilon$ combination, and report mean NLLs together with their 95% confidence intervals. Note that the error bars are plotted but are too small to see in the plot.

$$= \sum_{k_1=1}^{N_1} L_{1n_1k_1} L_{1m_1k_1} \cdots \sum_{k_D=1}^{N_D} L_{Dn_Dk_D} L_{Dm_Dk_D} \tag{58}$$

$$= \prod_{d=1}^{D} K_{dn_dm_d} \tag{59}$$

$$= k(x_{n_1\ldots n_D}, x_{m_1\ldots m_D}). \tag{60}$$

which is the required result. $\qquad\qquad\square$

## C  Additional results

### C.1  How effectively does the ConvCNP learn to undo the DP noise?

**Quantifying performance gaps.** In this section we provide some additional results on the performance of the DPConvCNP and the functional mechanism. Specifically, we investigate the performance gap between the DPConvCNP and the oracle (non-DP) Bayes predictor. Assuming the data generating prior is known, which in our synthetic experiments it is, the corresponding Bayes posterior predictive attains the best possible average log-likelihood achievable. We determine and quantify the sources of this gap in a controlled setting.

**Sources of the performance gaps.** Specifically, the performance gap can be broken down into two main parts: one part due to the DP mechanism (specifically the signal channel clipping and noise, and the density channel noise) and another part due to the fact that we are training a neural network to map the DP representation to an estimate of the Bayes posterior. To assess the performance gap introduced by each of these sources, we perform a controlled experiment with synthetic data from a Gaussian process prior (see Figure S1).

**Gap quantification experiment.** We fix the clipping threshold value at $C = 2.00$, which is a sensible setting since the marginal confidence intervals of the data generating process are $\pm 1.96$. We also fix the noise weighting at $t = 0.50$, which is again is a sensible setting since it places roughly equal

importance to the noise added to the density and the signal channels. We consider three different settings for the prior lenghtscale ($\ell = 0.25, 0.71, 2.00$) and two settings for the DP parameters ($\epsilon = 1.00, 3.00$ and fixed $\delta = 10^{-3}$). For each of the six combinations of settings, we train three different DPConvCNPs, one with just signal noise (red; no clip, no density noise), one with signal noise and clipping (orange; clip, no density noise) and one with signal noise and clipping and also density noise (green; clip, density noise). Note, only the last model has DP guarantees. We compare performance with the non-DP Bayesian posterior oracle (blue).

**Lower bound model.** When we only add signal noise to the ConvCNP representation (and do not apply clipping or add density noise), and the true generative process is a GP such as in this case, the predictive posterior (given the noisy signal representation and the noiseless density representation) is a GP. That is because the data come from a known GP, and the signal channel is a linear combination of the data (since we have turned off clipping) plus GP noise, so it is also a GP. Therefore, we can write down a closed form predictive posterior in this case. We refer to this as the *lower bound model* (black) in Figure S1, because for a given $C$ and $t$, the performance of this model is a lower bound to the NLL of any model that uses this representation as input. Note however that different lower bounds can be obtained for different $C$ and $t$.

**Conclusions.** We observe that the DPConvCNP with no clipping and no density noise (red) matches the performance of the lower bound model. This is encouraging as it suggests that the model is able to undo the effect of the signal noise perfectly. We also observe that applying clipping (orange) does not reduce performance substantially, which is also encouraging as it suggests that the model is able to cope with the effect of clipping on the signal channel, when it is trained to do so. Lastly, we observe that there is an additional gap in performance is introduced due to noise in the density channel (green). This is expected since the density noise is substantial and confounds the context inputs. This gap reduces as the number of context points increases, which is again expected. From the above, we conclude that in practice, the model is able to make predictions under DP constraints that are near optimal, i.e. there is likely not a significant gap due to approximating the mapping from the DP representation to the optimal predictive map, with a neural network.

### C.2   Supplementary model fits for the synthetic tasks

We also provide supplementary model fits for the synthetic, Gaussian and non-Gaussian tasks. For each task, we provide fits for three parameter settings ($\ell$ and $\tau$), two privacy budgets, four context sizes and two dataset random seeds. See Figures S2 to S5, at the end of this document, for model fits.

## D   Differentially-Private Sparse Gaussian Process Baseline

Here, we provide details of the differentially-private sparse variational Gaussian process (DP-SVGP) baseline.

Let $D = (\mathbf{x}, \mathbf{y})$ denote a dataset consisting of inputs $N$ inputs $\mathbf{x} \in \mathcal{X}^N$ and $N$ corresponding outputs $\mathbf{y} \in \mathcal{Y}^N$. We assume the observations are generated according to the probabilistic model:

$$f \sim \mathcal{GP}(0, k_{\theta_1}(x, x'))$$

$$\mathbf{y}|f, \mathbf{x} \sim \prod_{n=1}^{N} p_{\theta_2}(y_n|f(x_n)), \tag{61}$$

where $k_\theta$ denotes the GP kernel from which the latent function $f$ is sampled from, with hyperparameters $\theta$, and $\theta_2$ denotes the parameters of the likelihood function. Computing the posterior distribution $p_\theta(f|\mathbf{x}, \mathbf{y})$ is only feasible when the likelihood is Gaussian. Even when this is true, the computational complexity associated with this computation is $\mathcal{O}(N^3)$.

Sparse variational GPs [Titsias, 2009] offer a solution to this by approximating the true posterior with the GP

$$q_{\theta,\phi}(f) = p_\theta(f_{\neq \mathbf{u}}|\mathbf{u})q_\phi(\mathbf{u}) \tag{62}$$

with $\mathbf{u} = f(\mathbf{z})$, where $\mathbf{z} \in \mathcal{X}^M$ denote a set of $M$ inducing locations, and $q_\phi(\mathbf{u}) = \mathcal{N}(\mathbf{u}; \mathbf{m}, \mathbf{S})$. The computational complexity associated with this posterior approximation is $\mathcal{O}(NM^2)$, which is significantly lower if $M \ll N$. We can optimise the variational parameters $\phi = \{\mathbf{m}, \mathbf{S}, \mathbf{z}\}$ by

optimisation of the evidence lower bound, $\mathcal{L}_{\text{ELBO}}$:

$$\mathcal{L}_{\text{ELBO}}(\theta, \phi) = \mathbb{E}_{q_\theta(f)} \left[ \log p_\theta(\mathbf{y}|f(\mathbf{x})) \right] - \text{KL} \left[ q_\phi(\mathbf{u}) \| p_\theta(\mathbf{u}) \right]. \tag{63}$$

Importantly, $\mathcal{L}_{\text{ELBO}}$ also serves as a lower bound to the marginal likelihood $p_\theta(\mathbf{y}|\mathbf{x})$, and so we can optimise this objective with respect to both $\theta$ and $\phi$. Since the likelihood factorises, we can obtain an unbiased estimate to the $\mathcal{L}_{\text{ELBO}}$ by sampling batches of datapoints. This lends itself to stochastic optimisation using gradient based methods, such as SGD. By replacing SGD with a differentially-private gradient-based optimisation routine (DP-SGD), we obtain our DP-SVGP baseline.

A difficulty in performing DP-SGD to optimise model and variational parameters of the DP-SVGP baseline is that the test-time performance is a complex function of the hyperparameters of DP-SGD (i.e. maximum gradient norm, batch size, epochs, learning rate), the initial hyperparameters of the model (i.e. kernel hyperparameters, and likelihood parameters), and the initial variational parameters (i.e. number of inducing locations $M$). Fortunately, we are considering the meta-learning setting, in which we have available to us a number of datasets that we can use to tune these hyperparameters. We do so using Bayesian optimisation (BO) to maximise the sum of the $\mathcal{L}_{\text{ELBO}}$'s for a number of datasets. To limit the number of parameters we optimise using BO, we set the initial variational mean and variational covariance to the prior mean and covariance, $\mathbf{m} = \mathbf{0}$ and $\mathbf{S} = k(\mathbf{z}, \mathbf{z})$.

In Table S1, we provide the range for each hyperparameter that we optimise over. In all cases, we fix the number of datasets that we compute the $\mathcal{L}_{\text{ELBO}}$ for to 64 and the number of BO iterations to 200. We use Optuna [Akiba et al., 2019] to perform the BO, and Opacus [Yousefpour et al., 2021] to perform DP-SGD using the PRV privacy accountant.

| Hyperparameter | Min | Max |
|---|---|---|
| Max gradient norm | 1 | 20 |
| Epochs | 200 | 1000 |
| Batch size | 10 | 128 |
| Learning rate | 0.001 | 0.02 |
| Lengthscale | 0.1 | 2.5 |
| Period | 0.25 | 4.0 |
| Scale | 0.5 | 2.0 |
| Observation noise | 0.05 | 0.25 |

Table S1: The ranges of DP-SGD hyperparameter settings (upper half) and initial model hyperparameters (lower half) over which Bayesian optimisation is performed for the DP-SVGP baseline.

# E   Experiment details

In this section we give full details for our experiments. Specifically, we describe the generative processes we used for the Gaussian, non-Gaussian and sim-to-real tasks.

## E.1   Synthetic tasks

First, we specify the general setup that is shared between the Gaussian and non-Gaussian tasks. Second, we specify the Gaussian and non-Gaussian generative processes used to generate the outputs. Lastly we give details on the parameter settings for the amortised and the non-amortised models.

**General setup.** During training, we generate data by sampling the context set size $N \sim \mathcal{U}[1, 512]$, then sample $N$ context inputs uniformly in $[-2.00, 2.00]$ and 512 target inputs in the range $[-6.00, 6.00]$. We then sample the corresponding outputs using either the EQ Gaussian process or the sawtooth process, which we define below. For the DPConvCNP we use 6,553,600 such tasks with a batch size of 16 at training time, which is equivalent to 409,600 gradient update steps. We do note however that this large number of tasks, which was used to ensure convergence across all variants of the models we trained, is likely unnecessary and significantly fewer tasks (fewer than half of what we used) suffices. Throughout optimisation, we maintain a fixed set of 2,048 tasks generated

in the same way, as a validation set. Every 32,768 gradient updates (i.e. 200 times throughout the training process) we evaluate the model on these held out tasks, maintaining a checkpoint of the best model encountered thus far. After training, this best model is the one we use for evaluation. At evaluation time, we fix $N$ to each of the numbers specified in Figure 6, and sample $N$ context inputs uniformly in $[-2.00, 2.00]$ and 512 target inputs in the range $[-2.00, 2.00]$. We repeat this procedure for 512 separate tasks, and report the mean NLL together with its 95% confidence intervals in Figure 6. For all tasks, we set the privacy budget with $\delta = 10^{-3}$ and $\epsilon \sim \mathcal{U}[0.90, 4.00]$.

**Gaussian generative process.** For the Gaussian task, we generate the context and target outputs from a GP with the exponentiated quadratic (EQ) covariance, which is defined as

$$k(x, x') = \sigma_v^2 \exp\left(-\frac{(x-x')^2}{2\ell^2}\right) + \sigma_n^2.$$

**Sawtooth generative process.** For the non-Gaussian task, we generate the context and target outputs from a the truncation of the Fourier series of the sawtooth waveform

$$f(x) = \frac{2}{\pi} \sum_{m=1}^{2} \frac{\sin(2m\pi(dx/\tau) + \phi)}{m}$$

where $d \sim \mathcal{U}[-1, 1]$ is a direction sampled uniformly from $\{-1, 1\}$, $\tau$ is a period and $\phi \in [0, 2\pi]$ is a phase shift. In preliminary experiments, we found that the DPConvCNP worked well with the raw sawtooth signal (i.e. the full Fourier series) but the DP-SVGP struggled due to the discontinuities of the original signal, so we truncated the series, giving an advantage to the DP-SVGP.

**Non-amortised and amortised tasks.** For the non-amortised tasks, we train and evaluate a single model for a single setting of the generative parameter $\ell$ or $\tau$. Specifically, for the Gaussian tasks, we fix $\ell = 0.50, 0.71$ or $2.00$ and train a separate model for each one, that is then tested on data with the same value of $\ell$. Similarly, for the non-Gaussian tasks, we fix $\tau^{-1} = 0.25, 0.50$ or $1.00$ and train a separate model for each one, that is again then tested on data with the same value of $\tau$. For the amortised tasks, we sample the generative parameter $\ell$ or $\tau$ at random. Specifically, for the Gaussian tasks, we sample $\ell \sim \mathcal{U}[0.20, 2.50]$ for each task that we generate, and train a *single model* on these data. We then evaluate this model for each of the settings $\ell = 0.50, 0.71$ or $2.00$. Similarly, for the non-Gaussian tasks, we sample $\tau^{-1} \sim \mathcal{U}[0.20, 1.25]$ for each task that we generate, and train a *single model* on these data. We then evaluate this model for each of the settings $\tau^{-1} = 0.25, 0.50$ or $1.00$. The results of these procedures are summarised in Figure 6.

### E.2 Sim-to-real tasks

For the sim-to-real tasks we follow a training procedure that is similar to that of the synthetic experiments. During training, we generate data by sampling the context set size $N \sim \mathcal{U}[1, 512]$, then sample $N$ context inputs uniformly in $[-1.00, 1.00]$ and 512 target inputs in the range $[-1.00, 1.00]$. We then generate data by sampling them from a GP with covariance

$$k(x, x') = k_{3/2, \ell}(x, x') + \sigma_n^2, \tag{64}$$

where $k_{3/2, \ell}$ is a Matern-3/2 covariance with lengthscale $\ell \sim \mathcal{U}[0.50, 2.00]$ and noise standard deviation $\sigma_n \sim \mathcal{U}[0.30, 0.80]$. For all tasks, we set the privacy budget at $\delta = 10^{-3}$ and $\epsilon \sim \mathcal{U}[0.90, 4.00]$. The Dobe !Kung dataset is publicly available in TensorFlow 2 [Abadi et al., 2016], specifically the Tensorflow Datasets package. Note that we rescale the ages to be between $-1.0$ and $1.0$ and normalise the heights and weights of users to have zero mean and unit standard deviation. We assume that the required statistics for these normalisations are public, but they could be released with additional privacy budget. Inaccurate normalisations would only increase the sim-to-real gap and reduce utility, but not affect the privacy analysis. At evaluation time, we fix $N$ to each of the numbers specified in Figure 7. We then sample $N$ points at random from the normalised !Kung dataset and use the remaining points as the target set. We repeat this procedure for 512 separate tasks, and report the mean NLL together with its 95% confidence intervals in Figure 7.

### E.3 Optimisation

For all our experiments with the DPConvCNP we use Adam with a learning rate of $3 \times 10^{-4}$, setting all other options to the default TensorFlow 2 settings.

### E.4 Compute details

We train the DPConvCNP on a single NVIDIA GeForce RTX 2080 Ti GPU, on a machine with 20 CPU workers. Meta-training requires approximately 5 hours, with synthetic data generated on the fly. Meta-testing is performed on the same infrastructure, and timings are reported in Figure 5.

## F DPConvCNP architecture

Here we give the details of the DPConvCNP architecture used in our experiments. The DPConvCNP consists of a DPSetConv encoder, and a CNN decoder followed by a SetConv decoder. We specify the details for the parameters of these layers below.

**DPSetConv encoder and SetConv decoder.** For all our experiments, we initialise the DPSetConv and SetConv lengthscales (which are also used to sample the DP noise) to $\lambda = 0.20$, and allow this parameter to be optimised during training. For the learnable DP parameter mappings $t(\mu, N) = \mathtt{sig}(\mathrm{NN}_t(\mu, N))$ and $C(\mu, N) = \exp(\mathrm{NN}_C(\mu, N))$ we use simple fully connected feedforward networks with two layers of 32 hidden units each. For the discretisation step in the encoder, we use a resolution of 32 points per unit for all our experiments. We also use a fixed discretisation window of $[-7, 7]$ for the synthetic tasks and $[-2, 2]$ for the sim-to-real tasks. We did this for simplicity, although our implementation supports dynamically adaptive discretisation windows.

**Decoder convolutional neural network.** Most of the computation involved in the DPConvCNP happens in the CNN of the decoder. For this CNN we used a bare-bones implementation of a UNet with skip connections. This UNet consists of an initial convolution layer processes the signal and density channels, along with two constant channels fixed to the magnitudes $\sigma_s, \sigma_d$ of the DP noise used in these two channels, into another set of $C_\mathrm{in}$ channels. The result of the initial layer is then passed through the UNet backbone, which consists of $N$ convolutional layers with a stride of 2 and with output channels $C = (C_1, \ldots, C_N)$, followed by $N$ transpose convolutions again with a stride of 2 and output channels $C = (C_N, \ldots, C_1)$. Before applying each of these convolution layers, we create a skip connection from the input of the convolution layer and concatenate this to the output of the corresponding transpose convolution layer. Finally, we pass the output of the UNet through a final transpose convolution with $C_\mathrm{out} = 2$ output channels, which are then smoothed by the SetConv decoder to obtain the interpolated mean and (log) standard deviation of the predictions at the target points. For all our experiments, we used $C_\mathrm{in} = 32$, $N = 7$ and $C_n = 256$. We used a kernel size of 5 for all convolutions and transpose convolutions.

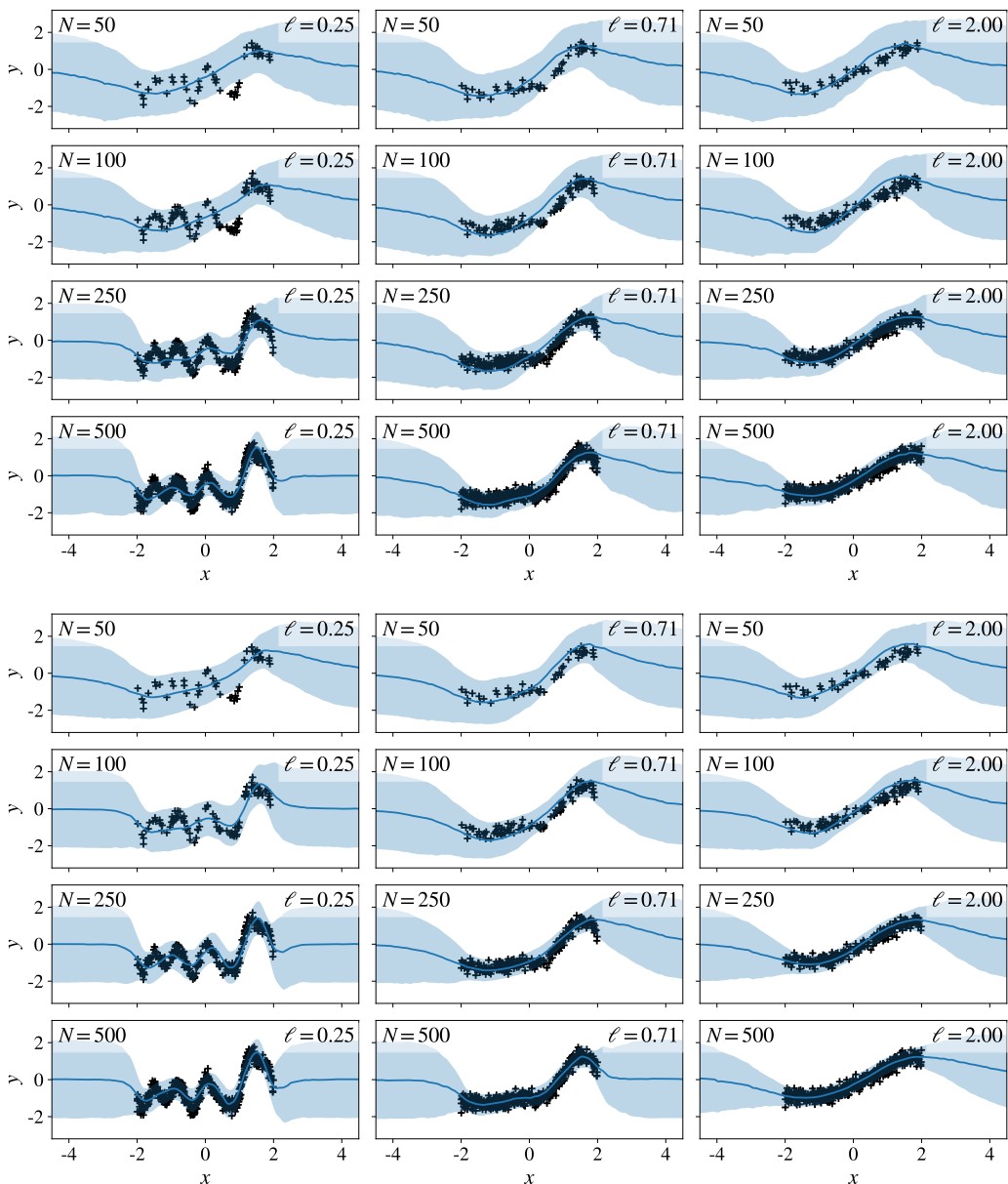

Figure S2: Example model fits for the DPConvCNP on the EQ GP task. For all the above fits, a single *amortised* DPConvCNP is used, that is a DPConvCNP that has been trained on EQ GP data with randomly chosen lengthscales $\ell \sim \mathcal{U}[0.20, 2.50]$ and random privacy budgets, specifically $\epsilon \sim \mathcal{U}[0.90, 4.00]$ and $\delta = 10^{-3}$. The first four rows correspond to $\epsilon = 1.00$ and the last four to $\epsilon = 3.00$. We have fixed $\delta = 10^{-3}$. Note that column-wise the datasets are fixed, and we are varying the context set size $N$.

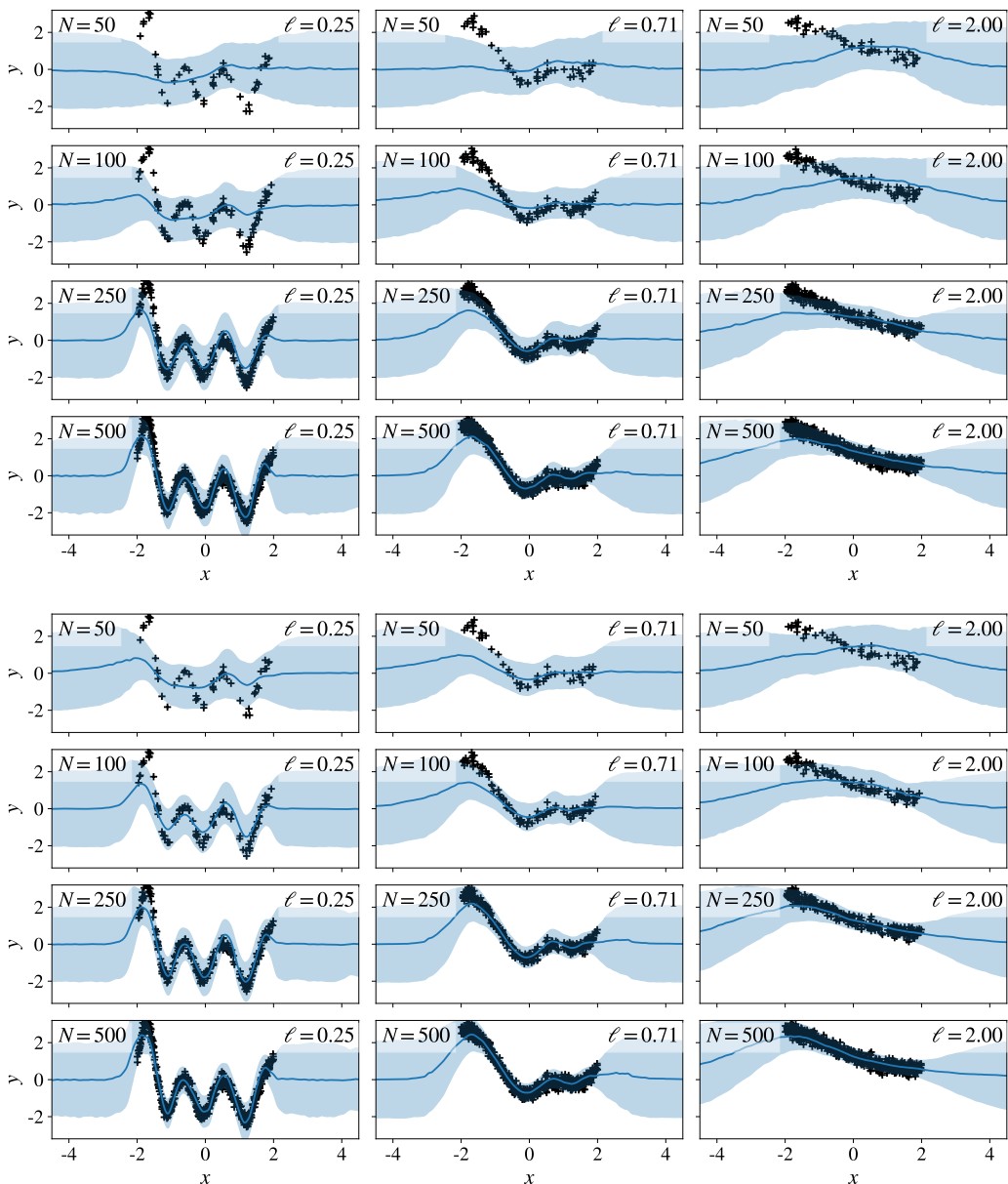

Figure S3: Same as Figure S2, but with a different dataset seed. Example model fits for the DPConvCNP on the EQ GP task. For all the above fits, a single *amortised* DPConvCNP is used, that is a DPConvCNP that has been trained on EQ GP data with randomly chosen lengthscales $\ell \sim \mathcal{U}[0.20, 2.50]$ and random privacy budgets, specifically $\epsilon \sim \mathcal{U}[0.90, 4.00]$ and $\delta = 10^{-3}$. The first four rows correspond to $\epsilon = 1.00$ and the last four to $\epsilon = 3.00$. We have fixed $\delta = 10^{-3}$. Note that column-wise the datasets are fixed, and we are varying the context set size $N$.

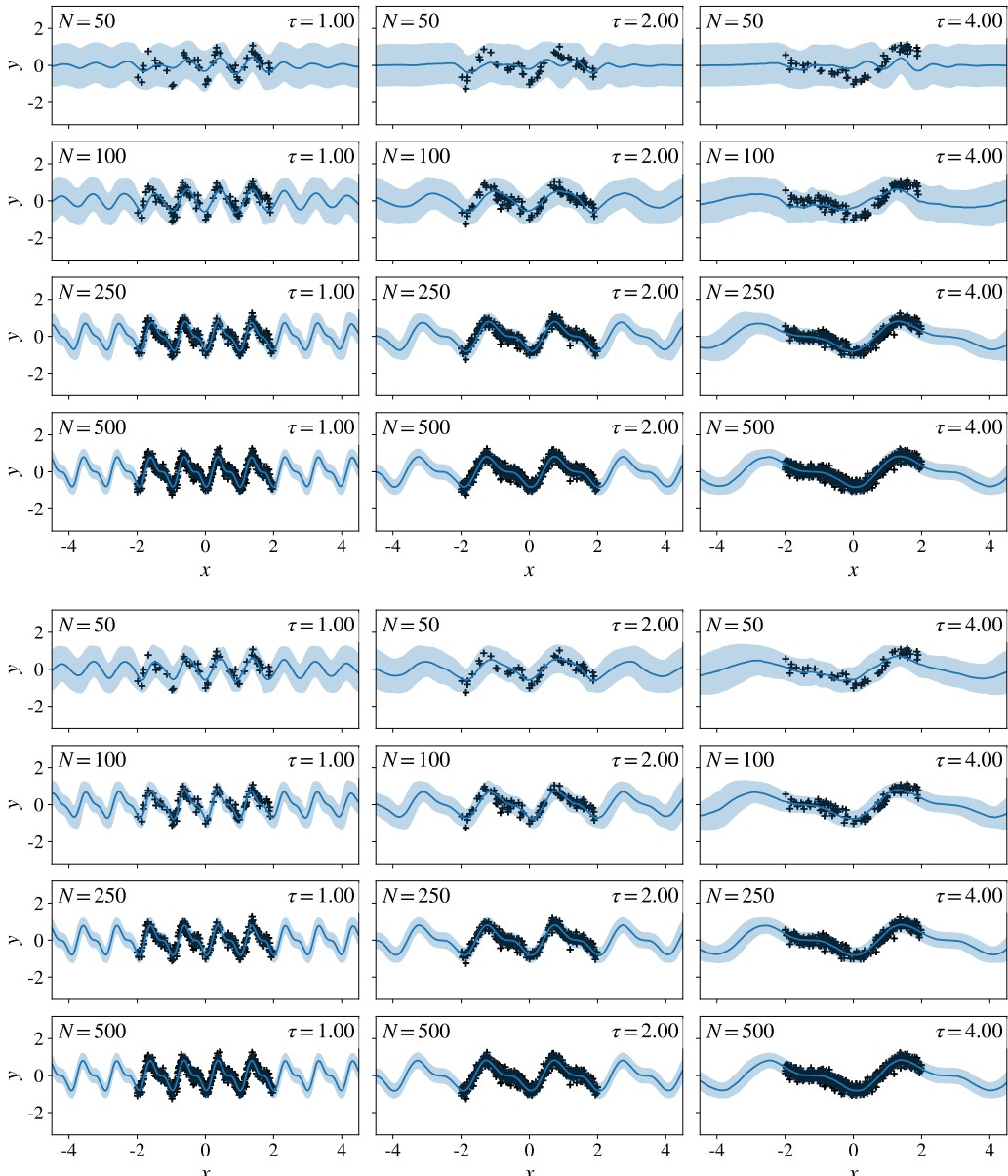

Figure S4: Example model fits for the DPConvCNP on the sawtooth task. For all the above fits, a single *amortised* DPConvCNP is used, that is a DPConvCNP that has been trained on sawtooth data with randomly chosen periods $\tau^{-1} \sim \mathcal{U}[0.20, 1.25]$ and random privacy budgets, specifically $\epsilon \sim \mathcal{U}[0.90, 4.00]$ and $\delta = 10^{-3}$. The first four rows correspond to $\epsilon = 1.00$ and the last four to $\epsilon = 3.00$. We have fixed $\delta = 10^{-3}$. Note that column-wise the datasets are fixed, and we are varying the context set size $N$.

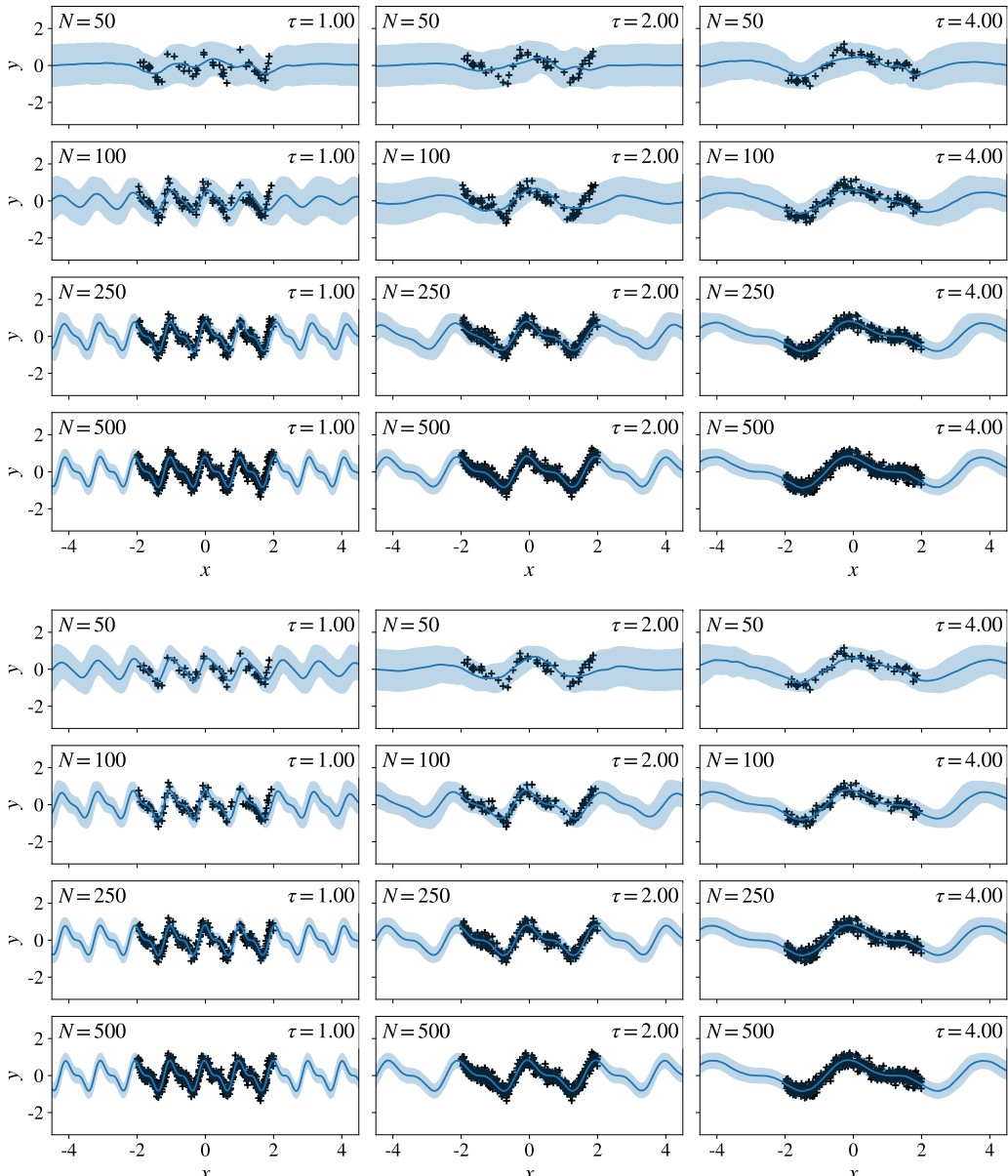

Figure S5: Same as Figure S4, but with a different dataset seed. Example model fits for the DPConvCNP on the sawtooth task. For all the above fits, a single *amortised* DPConvCNP is used, that is a DPConvCNP that has been trained on sawtooth data with randomly chosen periods $\tau^{-1} \sim \mathcal{U}[0.20, 1.25]$ and random privacy budgets, specifically $\epsilon \sim \mathcal{U}[0.90, 4.00]$ and $\delta = 10^{-3}$. The first four rows correspond to $\epsilon = 1.00$ and the last four to $\epsilon = 3.00$. We have fixed $\delta = 10^{-3}$. Note that column-wise the datasets are fixed, and we are varying the context set size $N$.

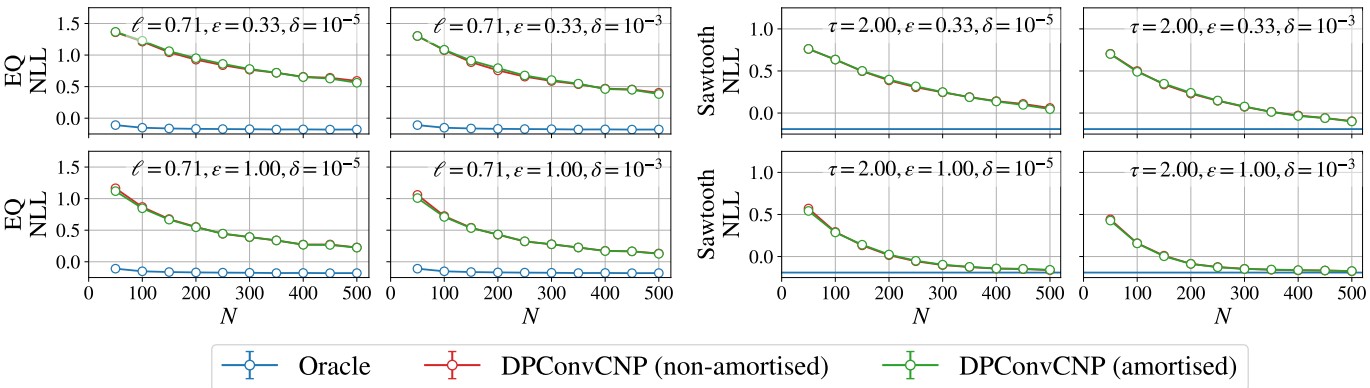

Figure S6: Additional results using the DPConvCNP on the EQ and sawtooth synthetic tasks with stricter DP parameters, namely all combinations of $\epsilon = \{1/3, 1\}$ and $\delta = \{10^{-5}, 10^{-3}\}$. The overall setup in this figure is identical to that in Figure 6, except the amortised DPConvCNP is trained on randomly chosen $\epsilon \sim \mathcal{U}[1/3, 1]$ and fixed $\delta = 10^{-5}$ or $10^{-3}$, and the non-amortised DPConvCNP models are trained on $\epsilon$ and $\delta$ values as indicated on the plots. Then, both amortised and non-amortised models are evaluated with the parameters shown on the plots. The DP-SVGP baseline was not run due to time constraints in the rebuttal period: it is significantly slower and more challenging to optimise than the DPConvCNP. We note that the amortisation gap, due to training a model to handle a continuous range of $\epsilon$ values, is negligible. We also note that as the number of context points $N$ increases, the performance of the DPConvCNP approaches that of the oracle predictors.

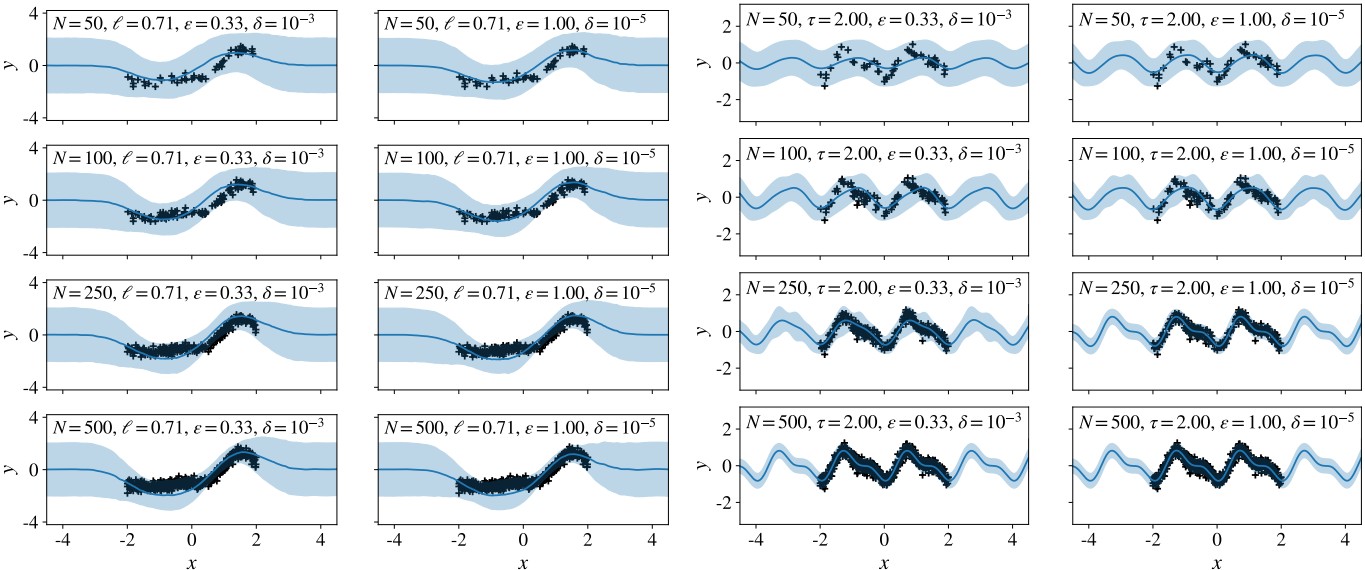

Figure S7: Illustrations of model fits on the synthetic EQ and sawtooth tasks, using stricter DP paramters, for different context sizes $N$. Left: model fits of amortised DPConvCNPs trained on EQ data using $\epsilon \sim \mathcal{U}[1/3, 1]$ and fixed $\delta = 10^{-3}$ (first column) or $\delta = 10^{-5}$ (second column) and evaluated on the DP parameters shown in the plots. Right: same as the left plot, except the data generating process is the sawtooth waveform rather than an EQ Gaussian process. We observe that the DPConvCNP produces sensible predictions even under strict privacy settings.

