# OpenReview forum: "Noise-Aware Differentially Private Regression via Meta-Learning"
_NeurIPS.cc/2024/Conference — NeurIPS 2024 poster_

### Official Review · Reviewer_aFHD · 2024-07-07

**Soundness:** 3
**Presentation:** 2
**Contribution:** 3
**Rating:** 6
**Confidence:** 4

**Summary:**

The paper proposes a meta learning gaussian dp algorithm. It uses the same framework as ConvCNP but replaces the encoder during train and meta test with a noisy encoder obtained by adding gaussian noise. The paper is not particularly well written and I have trouble following the notation in the main paper and appendix. I have significant doubts about the correctness of the method due to conflation between continuous results (the integral of a gaussian kernel) with discrete results (the sum of the kernel at discrete points).

Note: after feedback, the authors addressed my concerns about correctness.

**Strengths:**

The paper considers the use of meta learning to avoid an unfortunate trend in the dp literature which assumes the existence of large scale non private data with a similar distribution to a private dataset. Meta learning is a convenient possible alternative if one can create a good enough synthetic dataset.

Assuming everything is correct, the technique outperforms some prior work on simple synthetic and real datasets.

**Weaknesses:**

The experiments consider alternatives that use similar methods (GP or DP-SGD) but not alternatives that use different methods for similar problems (regression/kernel density estimates, clustering, etc.). I believe the technique would outperform the alternatives anyway, but their omission is noteworthy.

The experimental datasets are very simple and not comparable to the ones where public datasets are often assumed (images and NLP). Since the technical contribution lies in replacing r with a noisy r, there should be more challenging experiments and discussions on how to create simulated data for them.

A few typos:
- line 94, r should be theta
- line 130, missing word after predictive
- line 224 references eq 6 instead of 7

The delta value of 10^-3 is large compared to the literature. Setting it at 1/N puts it into the well-known privacy violating regime where one can return a random record with no noise, violating the privacy of someone with probability 1.

The paper seems to be taking too much credit for minor things, like using Gaussian DP as a drop-in replacement in the work of Hall et al. (literally taking a theorem that was meant as a drop-in replacement of prior formulas and using it as a drop-in replacement for those formulas). That is not a contribution.

The most important issue for last. I am not convinced that the algorithm is differentially private at the claimed parameter levels. The paper is missing a detailed pseudocode that puts everything together, but my understanding is that nothing changes except that wherever r is needed, a noisy version is used instead. The privacy properties should be provable from first principles (without relying on abstract kernel properties or an RKHS). Plug in the form of the kernel you are using and directly compute the sensitivity accounting for all of the discretized points being used. The reason I am asking for this is because the sensitivity calculation detailed in the appendix appears to be incorrect (it is not very well written and the notation is not explained well, so I am not 100% sure). The key to the proof seems to be that the gaussian kernel integrates to 1, but this is used to replace a discrete summation (i.e., \int_{-\infty}^\infty exp(-(x-y)^2))~dy  is very different from \sum_{i=1}^n exp(-(x-y_i)^2)). The summation can diverge depending on what y_i are chosen.
I would like to see a detailed privacy proof (with carefully explained notation) that explicitly uses the discretized point set (explaining exactly how the discretized points are chosen) and an explicitly defined covariance matrix.

**Questions:**

I would like to see a detailed privacy proof (with carefully explained notation) that explicitly uses the discretized point set (explaining exactly how the discretized points are chosen) and an explicitly defined covariance matrix.

For more challenging problems, such as text and images, how would one create simulated data so that DPConvCNP would be competitive?

**Limitations:**

The authors addressed limitations.

---

> ### Author Rebuttal · Authors · 2024-08-06
>
> > The delta value of 10^-3 is large compared to the literature. Setting it at 1/N puts it into the well-known privacy violating regime where one can return a random record with no noise, violating the privacy of someone with probability 1.
>
> This would be true if we considered any $(\epsilon, \delta)$-DP mechanism. However, for our specific mechanism, we can always adjust $\delta$, and obtain a different $\epsilon$, which will be finite, so we will not be in the privacy violating regime you describe. Making this adjustment would only change the $\epsilon$ values in our plots for DPConvCNP. We would also need to make the same adjustment for DP-SVGP, which will not be identical, since DP-SVGP is based on DP-SGD. However, we expect these differences to be small, so the adjustment would not change the conclusions from our results.
> To illustrate this, if we had used $\delta = 10^{-5}$ instead, which is the recommended value by NIST, we would adjust the DPConvCNP $\epsilon$ values we use according to the following table:
>
> | Inititial $\\epsilon$ | Adjusted $\\epsilon$ |
> | -------- | ------- |
> | 1.0 | 1.51 |
> | 3.0 | 4.20 |
>
> However, please also note that we have conducted additional experiments, attached to the pdf in the global response, which should help put your concerns about the value of $\\delta$ to rest.
>
> > The paper seems to be taking too much credit for minor things, like using Gaussian DP as a drop-in replacement in the work of Hall et al. (literally taking a theorem that was meant as a drop-in replacement of prior formulas and using it as a drop-in replacement for those formulas). That is not a contribution.
>
> We believe our formulation in the Introduction and the rest of the paper gives appropriate credit to prior work, but are happy to hear if you can suggest a better formulation. We also point out that both reviewers Eg3Y and AqSn found this contribution to be meaningful and listed it as a strength of our work.
>
> > The most important issue for last. I am not convinced that the algorithm is differentially private at the claimed parameter levels.
>
> We break down this paragraph and answer each of the points below.
>
> > my understanding is that nothing changes except that wherever r is needed, a noisy version is used instead.
>
> This is correct, but we also add clipping to the signal channel to bound sensitivity.
>
> > The sensitivity calculation detailed in the appendix appears to be incorrect…
>
> The only properties of the kernel we use in the sensitivity calculation are that $0 \leq k(x, y) \leq 1$ and $k(x, y) = k(y, x)$, which are looking at point evaluations, not integrals or sums. The sums in Eq. (19) and (27) disappear because the two datasets in the supremum only differ in one datapoint, so all but one term of the sums are zero.
>
> The transition from the functional form that the functional mechanism releases to the evaluations on the discretised point set follows from Proposition 5 of Hall et al. (2013). We will add a corollary to Theorem 4.2 that clarifies this.
>
> We would be happy to improve the notation if you could point out any specific part that is unclear.
>
> > The privacy properties should be provable from first principles… I would like to see a detailed privacy proof [...] that explicitly uses the discretized point set…
>
> Here is a proof that uses the Gaussian mechanism on the discretised points directly. Let $k(x_1, x_2) = \exp(-\frac{(x_1 - x_2)^2}{\lambda})$, and let the discretised points be arbitrarily chosen $x_1, \dotsc, x_m$. Let’s look at the density channel $r_d$ first. DPConvCNP computes the vector $[r_d(x_1), \dotsc, r_d(x_m)]$, and adds Gaussian noise with covariance $\sigma_d^2 M$, where $M$ is an $m\times m$ matrix with $M_{ij} = k(x_i, x_j)$. By Lemma A.4, it suffices to find the upper bound $\Delta$ in Eq. (14) for the matrix $M$ and the vector $r_d$ taking the role of the function $f$ of the lemma. By Proposition 8 of Hall et al. (2013), the sensitivity in Eq. (14) is bounded by the RKHS sensitivity we look at in Appendix A.4, which gives the privacy guarantee we claim. The privacy of the signal channel is proven in the same way, and releasing both of them is a composition.
>
> > For more challenging problems, such as text and images, how would one create simulated data so that DPConvCNP would be competitive?
>
> These domains are likely too high-dimensional for the ConvCNP model to be applied. More concretely, while ConvCNPs have been applied to image-based data before (see e.g. Gordon et al. (2020)) these applications have been confined to image in-painting rather than, for example, image classification. In image in-painting with meta-learning, the context set is a single partially observed image consisting of context pixels. The ConvCNP can be applied to this setting, but preserving data-point (i.e. pixel) privacy is meaningless. In image classification with meta-learning, the context set consists of several images, to which the ConvCNP cannot be readily applied. On the other hand, some amortized adaptation models (see Requeima et al. (2019)) have been developed, however these involve entirely different architectures and representations to the ConvCNP, so the functional mechanism does not apply. Overall, while very interesting and important, tackling higher dimensional data is well beyond the scope of this paper.
>
> Reference:
>
> James Requeima, Jonathan Gordon, John Bronskill, Sebastian Nowozin, Richard E. Turner, Fast and flexible multi-task classification using conditional neural adaptive processes, NeurIPS 2019.
>
> Thank you for your feedback. We hope we have settled your concerns regarding the validity of our theory. Provided our comments and amendments have satisfied your concerns, we would like to invite you to consider increasing your score.

---

> > ### Comment · Reviewer_aFHD · 2024-08-10
> >
> > Thank you for the clarifications. I am more confident in the correctness of the results and adjusted my score accordingly.

---

### Official Review · Reviewer_AqSn · 2024-07-12

**Soundness:** 4
**Presentation:** 3
**Contribution:** 4
**Rating:** 7
**Confidence:** 3

**Summary:**

This paper equips meta learning framework with DP guarantees. Specifically, datasets are split into context and target subsets respectively, where an encoder learns good representations from abundant context data, and is able to generalize to a limited amount of target data that may be sensitive. Since the output embedding (function) of the encoder is a sum of kernel functions indexed by the context dataset, so functional DP mechanisms are considered to protect the individual privacy in the dataset. Particularly, the authors extend the classical functional DP mechanism to Gaussian DP (GDP), which has theoretical benefits compared to alternatives. Experiments in both synthetic data and real data show its effectiveness.

**Strengths:**

1. Theoretically, the authors extend Gaussian DP to functional outputs, which may pave the way for follow-up works in related fields.
2. Empirically, the experiments are complete and extensive, i.e. on both synthetic and real tasks with comparison to existing works.
3. The paper is well-organized and easy to follow. Figure illustrations are clear and informative.

**Weaknesses:**

I am not familiar with the meta-learning framework (and related works), so it's likely I misunderstood some parts and had confusion about the meta-testing part.

Normally, a DP classifier, regressor, or generative model only introduces noise perturbation in the training (e.g. DP-SGD). Once the training is complete, the trained DP model can be applied to test data without worrying about breaching DP guarantees, as ensured by the post-processing theorem. In this work, however, the noise is injected in both training and test stages, where the authors explained that it accounts for mismatch between training and test. This paradigm does not look optimal to me. So here I have a few questions:
1. Can you explain why adding noise to the test stage is necessary? For example, compared to pretraining on the synthetic data then DP fine-tuning on private data, what is the benefit of the current method?
2. It looks to me that if we need to add noise in the test stage, we need to accumulate the total privacy budget $\epsilon$ as more test data are coming in. Is it true? If so, how do you determine the $\epsilon$ in the experiment (e.g. eps=1 or 3 in Fig 5)?

**Questions:**

1. I am not familiar with the neural process either. By reading the paragraphs starting at line 118, I don't see how it is different from a normal neural network. Particularly, how it is able to produce well-calibrated prediction, and how this is evaluated in the experimental results?
2. For the conditional NP, I don't see where is the conditional input, or what does the conditional mean here?
3. In Figure 4, I wonder how these lines are drawn. Do you use any analytical forms to calculate, or some libraries/packages to compute? Also, are they converted into the same $(\epsilon, \delta)$-DP notion? Because the $\epsilon$ in different DP notions are not the same (although they share the same notation).

---

> ### Author Rebuttal · Authors · 2024-08-06
>
> > I am not familiar with the meta-learning framework (and related works), so it's likely I misunderstood some parts and had confusion about the meta-testing part.
>
> Meta-learning has two phases, meta-training and meta-testing. You should view meta-testing as analogous to supervised learning: the input is a dataset, and the output is a function (model) $f$ that can make predictions at arbitrary inputs. Meta-training is the process of learning an algorithm which, when given a new dataset, produces a function (model) that can be queried at arbitrary inputs to make predictions. In the neural process literature, this is achieved using the encoder-decoder architecture of Eq. (1). With this understanding let us address your points below. We will also clarify these points in the revised paper.
>
> > Can you explain why adding noise to the test stage is necessary?
>
> Yes, here is why adding noise and clipping are necessary in the test stage. During the test stage, the DPConvCNP takes a previously unseen private dataset as its context set $D^{(c)}.$ (This would be the training data set in regular supervised learning.) Then, the DPConvCNP converts $D^{(c)}$ into a representation $r$ which is “published.” Once the representation is published, it can be passed through the rest of the architecture (the decoder) together with arbitrary test (target) inputs $x^{(t)}$ to make predictions for the corresponding test (target) outputs $y^{(t)}.$ If we do not apply noise and clipping in the test stage, then publishing $r$ would have no guarantees at all, and that could completely breach user privacy. By clipping and adding noise, we ensure that $r$ can be published with DP guarantees, and that arbitrarily many predictions (at arbitrary inputs) can be made using the decoder, without incurring any further privacy cost.
>
> We will add a corollary to Theorem 4.2 which states that Algorithm 2 with the DPSetConv encoder is DP with respect to meta-testing context set, in order to clarify precisely which part of the algorithm is DP, to better address this point.
>
> > [...] noise is injected in both training and test stages, where the authors explained that it accounts for mismatch between training and test. This paradigm does not look optimal to me.
>
> From our reply above, we hope it is clear why adding noise is necessary at test-time in order to preserve privacy. Let us now address meta-training time.
>
> At meta-training, we do not strictly need to apply noise or clipping, because the meta-training data are synthetic, so training the model on them does not incur a privacy cost. For example, we can train a standard ConvCNP model on synthetic data, and then replace its SetConv encoder with a DPSetConv encoder. The $r$ published from this model would still be $(\\epsilon, \\delta)$-DP at test time. However, applying noise and clipping changes the statistics of the representation $r$ in a way that this model is not trained for, so its predictions are catastrophically poor. Our observation is that we can address this issue by simply training the model with noise and clipping in place (i.e. with the DPSetConv) at meta-train time, teaching the model to account for these operations, and eliminating the mismatch. We will clarify which data is simulated and which is real in Algorithms 1 and 2.
>
> > It looks to me that if we need to add noise in the test stage, we need to accumulate the total privacy budget epsilon as more test data are coming in. Is it true?
>
> In our case, we assume that the meta-test context dataset is available all at once and does not change over time. This assumption is analogous to a supervised learning setting, where the training dataset (on which the supervised algorithm is applied) is fixed at the start. Just as in supervised learning, we can still make an arbitrary number of predictions at arbitrary locations, without any further privacy cost.
>
> It would be interesting to consider the scenario where more test data come in one by one or in batches, but this is beyond the scope of our current work.
>
>
> > For the conditional NP, I don't see where is the conditional input, or what does the conditional mean here?
>
> The term “conditional” neural process was introduced by Garnelo et al. (2018) to capture the fact that this meta-learning model models the conditional predictive distribution $p(y^{(t)} | x^{(t)}, D^{(c)}).$ We emphasize that we did not introduce this term, but rather the original authors of the CNP.
>
> > In Figure 4, I wonder how these lines are drawn. Do you use any analytical forms to calculate, or some libraries/packages to compute? Also, are they converted into the same (epsilon, delta)-DP notion? Because the epsilon in different DP notions are not the same (although they share the same notation).
>
> Yes, all of the noise levels in Figure 4 are calculated via closed form expressions and, if necessary, lightweight numerical solution routines. All of them use $(\epsilon, \delta)$-DP from Definiton 3.1
>
> In more detail, the classical line uses Theorem 3.7. The GDP line uses Definition 3.2 to convert the $(\epsilon, \delta)$-bound into a GDP $\mu$-bound by numerically solving $\mu$ from Eq. (5), and finds $\sigma$ with Theorem 4.1. For the RDP line, we get an RDP guarantee from Corollary 2 of Jiang et al. (2023), which we convert to $(\epsilon, \delta)$ with Proposition 3 of Jiang et al. (2023). RDP has the $\alpha$ parameter that can be freely chosen, so we optimise $\alpha$ to minimise $\epsilon$ for a given $\sigma$, and finally solve for $\sigma$, which gives the plotted values. Both the optimisation and finding $\sigma$ can be done analytically in this case.
>
> > I am not familiar with the neural process either. [...]
>
> Due to the limited character count of the rebuttal, we have answered this in a separate comment below.
>
> Thank you for your valuable feedback. If our comments and amendments have satisfied your concerns, we would like to invite you to consider increasing your score.

---

> ### Author Response · Authors · 2024-08-07
> **Additional Response**
>
> > I am not familiar with the neural process either. [...] how it is different from a normal neural network. [...] how it is able to produce well-calibrated prediction, and how this is evaluated in the experimental results?
>
> Here we give a brief answer to your question following the standard exposition from the literature (see e.g. Garnelo et al. (2018)). While a neural process is parameterised by neural networks (the encoder and the decoder) it differs from standard supervised neural network models in two main ways: the architecture and the training method.
>
> In terms of architecture, all neural processes involve an encoder-decoder architecture where (a) the encoder is designed to be invariant with respect to permutations of the context points in $D^{(c)},$ i.e. permuting order of the entries in the dataset $D^{(c)}$ leaves the output representation $r$ invariant; and (b) the decoder is constructed such that, given the representation $r,$, the predictions for each target $y^{(t)}_i$ depend only on the corresponding inputs $x^{(t)}_i$ and no other variable (see Eeq. (1)). These design choices are important for ensuring _Kolmogorov’s extension theorem_ is satisfied, in order to define a valid stochastic predictive process. We won’t delve further into this, but refer you to Kolmogorov’s extension theorem in Oksendal (2013) if you are further interested. For our purposes here, we can summarise this by saying that NPs have a particular architecture, chosen to satisfy Kolmogorov’s extension theorem.
>
> In terms of training, a neural process is again substantially different from a simple supervised neural network. A supervised network is trained on a single supervised dataset, and as such it is prone to overfitting. By contrast a neural process is trained on a (possibly infinite) collection of datasets. During meta-training (see Aalgorithm 1), the neural process is trained to make predictions for an unseen target set $D^{(t)}$ given an observed context set $D^{(c)}.$ Because $D^{(t)}$ are unseen, the model must learn to produce not just an accurate mean prediction, but also a sensible confidence interval for these.
>
> Our experimental results validate that the neural process predictions are well calibrated both qualitatively (e.g. see the calibrated confidence intervals in Ffigures S.2 to S.5 in the appendix) as well as quantitatively: in Figure 6 we see that the DPConvCNP performance approaches that of the perfectly calibrated oracle predictors, which can only happen if the DPConvCNP predictions are also well-calibrated. We stress that well-calibrated predictions have been demonstrated extensively in the neural process literature (see e.g. confidence intervals for the ConvCNP in Gordon et al. (2020)), and our results suggest the DPConvCNP also produces well-calibrated predictions.
>
> B. Oksendal, Stochastic differential equations: an introduction with applications (2013)

---

> > ### Comment · Reviewer_AqSn · 2024-08-12
> >
> > Thanks for the reply, I have raised my rating

---

### Official Review · Reviewer_Eg3Y · 2024-07-13

**Soundness:** 3
**Presentation:** 4
**Contribution:** 2
**Rating:** 7
**Confidence:** 4

**Summary:**

Authors propose DPConvCNP, a meta-learning model with a functional DP mechanism. This model is a modification to the SetConv procedure, applying clipping and work from [Hall et al. 2013] with a tighter Gaussian mech. analysis from [Dong et. al 2022] to conduct a sensitivity analysis and privatize the algorithm. The authors then demonstrate empirically that DPConvCNP provides both performance and efficiency boosts relative to the baseline of applying DP-SGD to a neural GP process on both some synthetic tasks and the !Kung dataset.

**Strengths:**

Overall: The paper follows a standard format for a contribution in differential privacy for machine learning: adapt an existing ML approach (SetConv) in a DP setting through sensitivity analysis and the application of a DP mechanism (in this case, an adapted version of the functional mechanism given by Hall et. al). Then, they show that this approach outperforms a Naive application of DP to some other standard method (in this case, applying DP-SGD to a standard neural implementation of a GP, which would be an obvious first attempt at the problem).

In the Appendix, I reviewed the proofs and lemmas for Theorems 4.1 and 4.2, they checked out and the analysis is well presented. I did not spend time checking Proposition B.1; it seems like the main trick there is noticing that you can apply linearity of expectation over the grid. It would be good if another reviewer has reviewed this.

S1. I find the quality of this submission’s presentation, from the intro to the supplementary appendix/proofs, to be exemplary. I commend the authors for the clarity of writing, notation, proofs and figures - it is refreshing, and worth highlighting. This is minus my minor nitpicks below, which will be easy to address.

S2. I find the approach intuitive, and the analysis sound. The experimental results are compelling, and this method is clearly extensible, as the authors hint at in their limitations section.

S3. The authors application of the [Dong et al.] result to the [Hall et al.] functional mechanism is indeed a nice contribution, and should be useful for future work.

**Weaknesses:**

I am deferring more substantial weaknesses to the questions section - broadly, I would like to raise my score, given adequate answers to my questions.

Nit: translation equivariant abbreviation TE should be introduced in the section where it is used heavily e.g. Section 3.2, not in the intro, along with a citation, which is confusing and gets lost for the reader.

Nit: in Section 5, line 259, “we make the GP variational…” is maybe missing a word or two.

Nit: In paragraph “Gradient based vs amortized…” in Section 3, the “on one hand…on the other hand” construction is difficult to follow, please restructure.

**Questions:**

Q1 : My main question/concern with the work is *how* important is leveraging a powerful hyperparameter tuning library (like BayesOpt from Optuna) to adjust hyperparameters for the empirical performance of the proposed method DPConvSet?

Having worked with DP-SGD variants (like Opacus’s private engine for optimizers), it’s clear to me that minor fluctuations in hyperparameters affect performance drastically. Recent work [Papernot et. al https://arxiv.org/pdf/2110.03620], [Mohapatra et. al, https://arxiv.org/pdf/2111.04906], and [Koskela et. al, https://arxiv.org/pdf/2301.11989] touches on the importance of private tuning. Can the authors discuss the delta between untuned and tuned versions of their algorithms, to be more “honest” about the effects of hyperparameter tuning on both their baseline and their method? An experiment (even anecdotal) would be nice, although isn’t necessarily required.

Q1.5 : Related in Q1, in section 4.3, you include privacy parameters in the meta training of DPConvCNP (as far as I can tell) by re-parameterizing. Can you explain how this maintains the ($\epsilon,\delta$)-DP guarantee if the iterative meta-learning procedure is conditioned on prior trainings?

Q2 : It seems like a missed opportunity to discuss the shortcomings of standard supervised learners on small datasets, even a simple standard private regression (many open source implementations available, for example https://diffprivlib.readthedocs.io/en/latest/modules/models.html#linear-regression). Perhaps comparisons between standard learners and meta learners is not worth it in the data scenarios you explore? It’d be helpful for me if you could discuss this (a light experiment if appropriate).

Q3 : Can the authors justify their choice of privacy hyperparameters in the text? I acknowledge that $\epsilon$ of 1.0+ is definitely reported on in the literature and used practically, but it’s good to contrast this with  $\epsilon < 1.0$, as this is the more theoretically comfortable private regime, in a strict sense. Adding results for this wider range of privacy parameters would strengthen the contribution.

**Limitations:**

The authors do a very nice job of highlighting the limitations of their work, alongside broader impacts. This is greatly appreciated!

---

> ### Author Rebuttal · Authors · 2024-08-06
>
> > Q1 : My main question/concern with the work is how important is leveraging a powerful hyperparameter tuning library (like BayesOpt from Optuna) to adjust hyperparameters for the empirical performance of the proposed method DPConvSet?
>
> We only use Optuna and BayesOpt on the DP-SVGP baseline. We did not perform any BayesOpt tuning on any hyperparameters of the DPConvCNP.
>
> Instead, rather than performing BayesOpt, the DPConvCNP “tunes” the parameters (but no hyperparameters such as model size) of the backbone network as well as the NNs for parameterising the DP clipping threshold $C$ and noise weight $t$ (Section 4.3) by performing gradient descent on synthetic data. We found our procedure to work well out-of-the-box and did not make any further efforts to optimize the DPConvCNP.
>
> > Q1.5 : Related in Q1, in section 4.3, you include privacy parameters in the meta training of DPConvCNP (as far as I can tell) by re-parameterizing. Can you explain how this maintains the (epsilon, delta)-DP guarantee if the iterative meta-learning procedure is conditioned on prior trainings?
>
> During meta-training, DPConvCNP only uses simulated data, so meta-training does not have any impact on the privacy of the real data. We include the noise addition, clipping and other privacy-related computations during meta-training in order to make the model learn the same task it will do during meta-testing, either for single privacy parameters or over a range of privacy parameters. In turn, at meta-testing, the DPSetConv ensures that the data representation of the private data is $(\\epsilon, \\delta)$-DP.
>
> > Recent work [...] touches on the importance of private tuning. Can the authors discuss the delta between untuned and tuned versions of their algorithms, to be more “honest” about the effects of hyperparameter tuning on both their baseline and their method?
>
> Both the DPConvCNP and the DP-SVGP are trained on simulated data without any privacy cost, so the work on private hyperparameter tuning is not relevant to our setting.
>
> > Q2: It seems like a missed opportunity to discuss the shortcomings of standard supervised learners on small datasets, even a simple standard private regression [...]. Perhaps comparisons between standard learners and meta learners is not worth it in the data scenarios you explore?
>
> Standard DP supervised learning algorithms would likely not work very well in our settings. For example, linear regression, which seems to be the only regression algorithm in the library you linked, would clearly need some nonlinear features to have any chance of working in our settings. Designing appropriate nonlinear features that work well with DP would be a non-trivial effort, to the point that we would effectively be designing a new baseline. In non-private settings, GPs are the state-of-the-art in the low-data regime, so we think the DP-SVGP is the most reasonable baseline we could use. We will further explain our choice of baseline in the revised version.
>
> > Q3: Can the authors justify their choice of privacy hyperparameters in the text? [...] Adding results for this wider range [$\\epsilon < 1$] of privacy parameters would strengthen the contribution.
>
> As you acknowledge in your question, our privacy parameter settings are standard in the literature. Relatively few papers go below $\\epsilon = 1.$ However, following your advice we have run some further experiments for $\\epsilon < 1,$ as well as smaller $\\delta.$ We have attached these results in the additional rebuttal pdf of the "global response." There, we observe that the DPConvCNP performs well even in such stricter privacy settings, given enough data.
>
> Thank you for your valuable feedback. If our comments and amendments have satisfied your concerns, we would like to invite you to consider increasing your score.

---

> > ### Comment · Reviewer_Eg3Y · 2024-08-07
> >
> > I thank the authors for their clarifications and additional experiments at lower $\epsilon$ values, the effort is appreciated. Specifically, I feel as though their responses have fully helped me understand how their method preserves privacy; this indicates to me that they will be able to make clarifying changes to their paper were it accepted. I have a further clarification, which I hope the authors' can help me with.
> >
> > > During meta-training, DPConvCNP only uses simulated data, so meta-training does not have any impact on the privacy of the real data. We include the noise addition, clipping and other privacy-related computations during meta-training in order to make the model learn the same task it will do during meta-testing, either for single privacy parameters or over a range of privacy parameters. In turn, at meta-testing, the DPSetConv ensures that the data representation of the private data is
> > -DP.
> >
> > Maybe I should have been more specific: is the simulated data making assumptions (e.g. that this information is public) about the domain/range of the input data it is simulating? Or is it inferring the domain/range directly? Or does it just use arbitrary data? Whatever the scheme, this needs to be made more clear; if the assumption is not that this information was public, for example, there would need to be a DP range estimation embedded in the algorithm.

---

> > > ### Author Response · Authors · 2024-08-09
> > >
> > > Thank you for your response. We are happy you have found our clarifications useful. With regards to your questions:
> > >
> > > > is the simulated data making assumptions (e.g. that this information is public) about the domain/range of the input data it is simulating? Or is it inferring the domain/range directly? Or does it just use arbitrary data? Whatever the scheme, this needs to be made more clear;
> > >
> > > The simulated meta-training data should ideally have similar statistics as the real data: the closer these statistics are, the better the sim-to-real approach will work. This can be achieved, for example, by generating diverse enough synthetic data to ensure that some of the meta-training datasets have similar statistics to the real data. Importantly however, the simulating generative process has to be picked without looking at the real data to avoid privacy leakage.
> > >
> > > With the Dobe !Kung data, we normalise the data so that age is in [-1, 1] and height and weight have zero mean and unit variance. We assume that the required statistics for these normalisations are public. In case the statistics were not public, they could easily be released with additional privacy budget. Inaccurate normalisations would only increase the sim-to-real gap and reduce utility, not affect the privacy analysis. The simulator produces the normalised data, and we picked its hyperparameters based on rough estimates of the weights, heights and ages of individuals we might expect to see in reality.
> > >
> > > We appreciate your point that the paper would benefit from a lengthier explanation of these nuances. We can easily fit this in the space afforded by the camera-ready version. We hope the above helps clarify any remaining questions you might have had about the paper. Please let us know if you’d like any further clarification.

---

> > > > ### Comment · Reviewer_Eg3Y · 2024-08-09
> > > >
> > > > Yes, this is the clarification I was looking for, thank you. Please specify this in the final version of the paper. I suspect that the performance of this method is sensitive to performing normalization based on statistics of the data that, if noisy (i.e. if actually *privately* measured), would affect the performance of the method to a significant degree (say, if you withheld 20% of your privacy budget to estimate the first 2 moments and the range of each continuous feature). That said, I'm satisfied so long as something like your point above is included in the final paper, and agree it doesn't affect the overall privacy analysis.
> > > >
> > > > I have raised my score accordingly - good luck!

---

### Author Rebuttal · Authors · 2024-08-06

Some reviewers had questions regarding how meta-learning differs from standard supervised learning, and the consequences of these differences from the point of view of privacy. To summarise, meta-learning has two phases, meta-training and meta-testing. Meta-testing is analogous to training in supervised learning: the input is a dataset, and the output is a function (model) that can make predictions at arbitrary inputs. Meta-training is the process of learning an algorithm which, when given a new dataset, produces a function (model) that can be queried at arbitrary inputs to make predictions.

We assume that the data used during meta-training comes from a simulator, so using it has no implications on privacy. On the other hand, we consider the meta-test data to be private, and design DPConvCNP to guarantee its privacy. Since the model from meta-testing is made private, we can use the model to make an arbitrary number of predictions.

We will make changes in the revision to further clarify these points, especially the distinction between private and simulated data. In particular, we will update the descriptions of Algorithms 1 and 2 in the following way to make it clear whether their input data is simulated or private (added words in italics):

**Algorithm 1**

**Input:** *Simulated* datasets $(D\_m)\_{m=1}^{M}$, encoder $\mathrm{enc}\_\phi$, decoder $\mathrm{dec}\_\theta$, iterations $T$, optimiser $\mathrm{opt}$.

**Algorithm 2**

**Input:** *Real* context $D^{(c)}$, $\mathrm{enc}\_\phi$, $\mathrm{dec}\_\theta$

We will also add the following corollary to Theorem 4.2 to make it clear which parts of DPConvCNP are private:

**Corollary.** Algorithm 2 with the DPSetConv encoder from Algorithm 3 is $(\epsilon, \delta)$-DP with respect to the real context set $D^{(c)}$.

Lastly, we would like to raise to the reviewers’ attention, particularly reviewers Eg3Y and aFHD, that we ran additional experiments with stricter DP $(\\epsilon, \\delta),$ following their advice that this would strengthen our manuscript. Our results show that the DPConvCNP produces sensible predictions even in strict privacy settings, given enough data.

---

### Decision · Program_Chairs · 2024-09-25

**Decision:**

Accept (poster)

**Comment:**

This paper proposes a meta learning algorithm that uses Gaussian DP and the ConvCNP framework. This approach is compared against the naive application of DP and shown to outperform it. Overall the reviewers agree that the analysis is sound, that the empirical results show an important improvement, and that the techniques/contributions may have broader use. This looks like a solid poster contribution. Despite this, I encourage the authors to look at presentation changes that can make understanding and analyzing the theoretical contributions easier.